# EZ Clear for simple, rapid, and robust mouse whole organ clearing

Chih-Wei Hsu[1,2,3,4]*, Juan Cerda III[1], Jason M Kirk[2], Williamson D Turner[1], Tara L Rasmussen[1,4], Carlos P Flores Suarez[1], Mary E Dickinson[1,2,4], Joshua D Wythe[1,4,5]*

[1]Department of Integrative Physiology, Baylor College of Medicine, Houston, United States; [2]Optical Imaging and Vital Microscopy Core, Advance Technology Cores, Baylor College of Medicine, Houston, United States; [3]Department of Education, Innovation and Technology, Baylor College of Medicine, Houston, United States; [4]Cardiovascular Research Institute, Baylor College of Medicine, Houston, United States; [5]Department of Neurosurgery, Baylor College of Medicine, Houston, United States

**Abstract** Tissue clearing for whole organ cell profiling has revolutionized biology and imaging for exploration of organs in three-dimensional space without compromising tissue architecture. But complicated, laborious procedures, or expensive equipment, as well as the use of hazardous, organic solvents prevent the widespread adoption of these methods. Here, we report a simple and rapid tissue clearing method, EZ Clear, that can clear whole adult mouse organs in 48 hr in just three simple steps. Samples stay at room temperature and remain hydrated throughout the clearing process, preserving endogenous and synthetic fluorescence, without altering sample size. After wholemount clearing and imaging, samples processed with EZ Clear can be subjected to downstream applications, such as tissue embedding and cryosectioning followed by standard histology or immunofluorescent staining without loss of fluorescence signal from endogenous or synthetic reporters. Furthermore, we demonstrate that wholemount adult mouse brains processed with EZ Clear can be successfully immunolabeled for fluorescent imaging while still retaining signal from endogenous fluorescent reporters. Overall, the simplicity, speed, and flexibility of EZ Clear make it easy to adapt and implement in diverse imaging modalities in biomedical research.

*For correspondence:
loganh@bcm.edu (C-WH);
wythe@bcm.edu (JDW)

Competing interest: The authors declare that no competing interests exist.

## Editor's evaluation

The manuscript reports a new tissue clearing procedure that is faster (clearing within 48 hours), uses less hazardous chemicals, and importantly appears to result in less tissue volume change compared to other methods. The simple protocol adds further to the toolbox of tissue clearing methods and is one that is likely to be even more popular than many current methods.

## Introduction

Over the past 30 years, the development of confocal microscopes that can image large samples at cellular resolution, combined with powerful increases in computing and the ability to handle large volumes of data, have unleashed an explosion in three-dimensional (3D) visualization of organ structures at both a macro and cellular scale (*Richardson and Lichtman, 2015*; *Richardson et al., 2021*). A key to this revolution has been the simultaneous deluge of tissue clearing protocols driven by advances in optical physics and chemical engineering (*Richardson and Lichtman, 2015*; *Richardson et al., 2021*; *Tainaka et al., 2016*). The development of both organic solvent-based clearing methods

like BABB (*Dodt et al., 2007*), 3DISCO (*Ertürk et al., 2012*; *Ertürk et al., 2011*), iDISCO (*Renier et al., 2014*), Ethanol-ECi (*Klingberg et al., 2017*), PEGASOS (*Jing et al., 2018*), Fast 3D (*Kosmidis et al., 2021*), and aqueous-based techniques such as Sca*le* (*Hama et al., 2011*; *Hama et al., 2015*), CLARITY (*Chung and Deisseroth, 2013*), PACT-PARS (*Yang et al., 2014*), CUBIC (*Susaki et al., 2014*), and Ce3D *Li et al., 2017* have been successfully leveraged in various biological model systems. As the depth of light microscopy-based imaging of large samples is usually limited by the scattering of light, advances in tissue clearing allows researchers to examine tissues in their native 3D state by imaging modalities with different volume capacities and resolution, ranging from optical projection tomography, to confocal or multiphoton imaging, to cutting edge lightsheet fluorescence microscopy (LSFM)-based imaging (*Udan et al., 2013*; *Hoog et al., 2018*; *Ahrens et al., 2013*; *Amat et al., 2015*; *Ueda et al., 2020*; *Kolesová et al., 2016*).

Although each of the aforementioned strategies has their own unique merits (*Richardson and Lichtman, 2015*; *Richardson et al., 2021*; *Tainaka et al., 2016*), several hurdles remain for the field in terms of developing and adopting a rapid, simple, and robust tissue clearing method. While aqueous-based protocols may preserve fluorescence from endogenous transgenic reporters, and are generally compatible with most existing imaging platforms, they often require either extended incubation periods (days to weeks) or complicated, laborious procedures, as well as special clearing equipment (*Richardson et al., 2021*; *Tainaka et al., 2016*; *Ariel, 2017*). For example, hydrogel scaffolding-based CLARITY clearing provides a robust and controllable workflow to clear tissue in an aqueous environment, but the complicated steps and considerable technical investment required represent substantial hurdles that prevent many researchers from adopting this methodology (*Chung and Deisseroth, 2013*). Moreover, as hydrogel scaffolding is based on covalently conjugating proteins in a polyacrylamide matrix, precise control of tissue fixation with paraformaldehyde (PFA) and thermal crosslinking of polyacrylamide is required. Otherwise, the overall strength and pore size of the hydrogel-tissue matrix varies experiment to experiment during SDS-mediated electrophoretic removal of lipids, limiting reproducible, robust results (*Tainaka et al., 2016*; *Chung and Deisseroth, 2013*; *Gradinaru et al., 2018*). In addition, variations in clearing efficiency for different organ systems based on tissue composition and volume also pose a significant challenge for researchers, as extensive optimization is required to tailor this method to individual research projects.

In contrast, organic solvent-based clearing methods are simple, fast, efficient, and do not require specialized homemade or commercial equipment. However, challenges in sample handling and imaging tissues in these hazardous, toxic solvents, combined with the rapid decline of endogenous fluorescent signal, limit their applicability (*Richardson and Lichtman, 2015*; *Tainaka et al., 2016*; *Ueda et al., 2020*; *Ariel, 2017*; *Gradinaru et al., 2018*). These clearing strategies typically use corrosive and combustible organic solvents to match the refractive index (RI) of dehydrated tissues to minimize light scattering and render tissue optically transparent, which limits downstream imaging to microscope systems with solvent-resistant chambers and objectives. Critically, while samples can be processed for conventional histology and immunostaining following aqueous-based tissue clearing (*Neckel et al., 2016*), whether solvent cleared samples can be processed for downstream techniques such as cryosectioning and immunolabeling (or even conventional histology) has not been rigorously explored, potentially limiting the application of these approaches to non-precious tissue samples.

Here, we present a simple and rapid tissue clearing method, EZ Clear, that effectively renders whole adult mouse organs optically transparent in 48 hrs using three simple steps: lipid removal, washing, and RI matching. EZ Clear combines the advantages of solvent-mediated lipid removal with highly water-miscible tetrahydrofuran (THF) and renders samples transparent in an aqueous, high RI (n=1.518) sample mounting and imaging solution that is compatible with most microscopy platforms. Because samples are submerged in an aqueous environment throughout the entirety of the protocol, no significant changes in sample size occur during tissue processing. Our results demonstrate that EZ Clear processed adult mouse organs (brain, eye, heart, lung, liver, spleen, kidney, testis, and ovary) not only retained endogenous fluorescence from transgenic reporters to allow 3D whole organ lightsheet imaging, but could also be successfully wholemount immunofluorescently stained while preserving endogenous fluorescent signal. Moreover, samples treated with EZ Clear can also be processed after wholemount imaging for cryosectioning and histology or immunofluorescent staining for further analysis. In summary, EZ Clear is a simple, robust, and easy to adopt whole organ clearing technique that can be applied to various sample volumes and utilized across most common imaging platforms.

## Results

### Clear mouse organs in 48 hr with three simple steps

EZ Clear effectively clears adult mouse organs in 48 hr with three simple steps at room temperature (*Figure 1A*). Step 1 is to immerse the fixed sample in the lipid removal solution, which consists of 50% (v/v) THF in sterile Milli-Q water. This formula allows THF to dissolve lipid away from the tissue while the organ remains hydrated in an aqueous environment. Following lipid removal, the sample is incubated in sterile Milli-Q water for 4 hr to wash and remove any residual THF from the tissue. The tissue is then rendered transparent by immersing it in aqueous RI matching and imaging solution (EZ View, RI = 1.518) at room temperature for 24 hr. EZ Clear not only renders the whole brain as transparent as 3DISCO and FAST 3D (*Figure 1B*, *Figure 1—figure supplement 1A*), but it also has the shortest processing time (48 hr) and simplest procedure (three steps) of current tissue clearing protocols. Because the sample is maintained in an aqueous environment throughout the entire process, the size of processed samples remains constant throughout the protocol (*Figure 1C*, n=4 for each condition, one-way ANOVA, size change ratio = 1.072 ± 0.062). Conversely, other approaches either significantly shrank (Fast 3D: 0.776±0.025 and 3DISCO: 0.593±0.013) or expanded (X-CLARITY: 1.608±0.049) the brain (*Figure 1C* and *Figure 1—figure supplement 1B*). In addition, endogenous transgenic fluorescent reporter activity is preserved throughout the EZ Clear protocol, as a brain from a *Thy1-EGFP* neuronal reporter line that was processed with EZ Clear retained robust signal within neurons when imaged by lightsheet microscopy across a total imaging depth of 5 mm (*Figure 1D–F* and *Video 1*). EZ Clear is also compatible with adult mouse organs other than the brain, with excellent transparency evident following tissue processing (*Figure 1G*). Multiple adult mouse organs (eye, heart, kidney, testis, and ovary) were collected after vascular perfusion with a far-red fluorescently conjugated *Lycopersicon esculentum* (tomato) lectin (lectin-DyLight 649) dye that labels the endothelium. Following perfusion, samples were fixed and then processed with EZ Clear. Wholemount imaging of samples by LSFM demonstrated that the lectin-649 signal was robust and well preserved (*Figure 1H*, maximum intensity projection). These results show that with three simple steps in 48 hr, EZ Clear effectively renders adult mouse organs optically transparent while simultaneously preserving signal from synthetic and endogenous fluorescent reporters.

### Tissue RI matching with aqueous EZ View sample mounting and imaging solution

To maintain cleared samples in an aqueous environment after lipid removal and washing, we compared the ability of various sample mounting and imaging buffers to render tissue optically transparent. Given the aqueous nature of EZ Clear, we first tested whether refractive index matching solution (RIMS) (80% (v/v) Nycodenz, RI = 1.46) (*McCreedy et al., 2021*), or the more affordable alternative sRIMS (80% D-sorbitol, RI = 1.43) (*Yang et al., 2014*), were compatible with tissues following lipid removal by EZ Clear. Somewhat surprisingly, EZ Clear delipidated samples immersed in either RIMS (*Figure 2C*) or sRIMS (data not shown) were not rendered transparent. Thus, we pursued novel aqueous solutions with an RI over 1.50. Previous studies showed that urea can hyperhydrate samples and enhance tissue transparency (*Hama et al., 2011*; *Susaki et al., 2014*; *Zhu et al., 2019*). We also noticed that the RIs of aqueous solutions increased linearly as the concentration of urea was increased (*Figure 2—source data 1*). To test the effect of combining urea and Nycodenz on RI and clearing efficiency under aqueous conditions, we gradually increased the urea concentration in RIMS and measured the RI (*Figure 2A*). As predicted, the RI of the 80% Nycodenz solution was positively correlated with the concentration of urea. While the solution was saturated with 8 M of urea in 80% Nycodenz, the highest RI (RI = 1.518 ± 0.0003) was obtained with 7 M urea in 80% Nycodenz at room temperature. We then examined sample transparency by comparing brains after lipid removal with EZ Clear, immersed and equilibrated in either PBS (RI = 1.332), RIMS (80% Nycodenz, RI = 1.463 ± 0.0019), or EZ View (80% Nycodenz and 7 M urea, RI = 1.518 ± 0.0003) (*Figure 2B–D*). While the tissue remained opaque in PBS (*Figure 2B*), the transparency of the EZ Clear lipid removed brain showed only a minor improvement following immersion and equilibration in RIMS for 24 hr (*Figure 2C*). However, the EZ Clear lipid removed brain immersed and equilibrated in EZ View solution for 24 hr was comparable to samples cleared using 3DISCO (compare *Figure 2D* and *Figure 1B*).

Next, we examined how equilibrating samples in different RIMS following EZ Clear lipid removal affected imaging depth. Adult mice perfused with fluorescent lectin-649 were euthanized and the

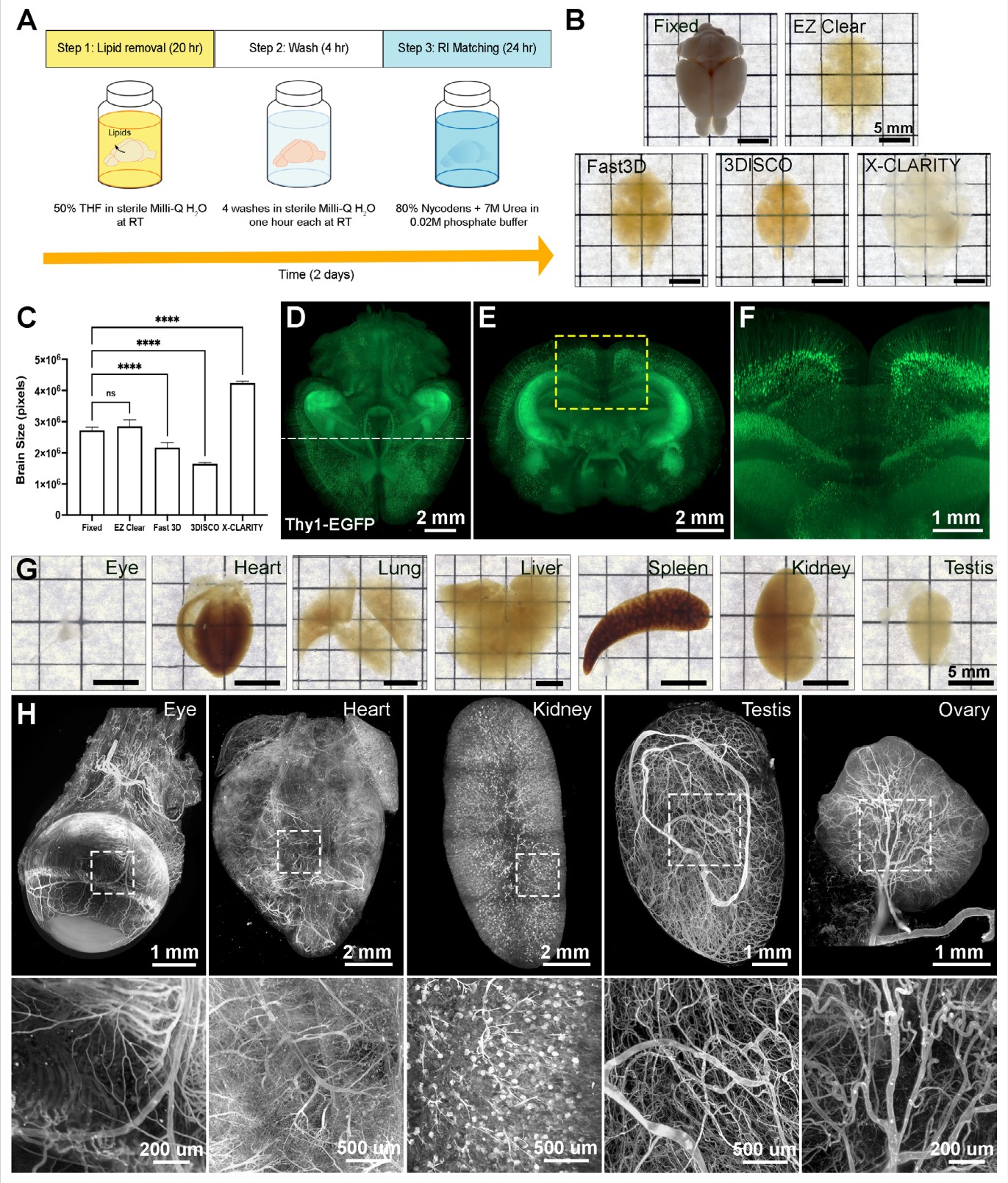

**Figure 1.** EZ Clear is a simple, rapid, and efficient tissue clearing process. (**A**) The three steps, requiring 48 hr of the EZ Clear tissue clearing protocol. (**B**) Brightfield representative images of adult mouse brains (9 weeks of age) fixed in 4% paraformaldehyde (PFA) only, or fixed and cleared using EZ Clear, Fast 3D, 3DISCO, or X-CLARITY. (**C**) Quantitative comparison of the volume changes of the mouse brains before and after processing with different clearing protocols. (n=4, one-way ANOVA. Error bars represent standard deviation (SD). ns – not significant, ****p<0.0001.) (**D–F**) Volume

*Figure 1 continued on next page*

*Figure 1 continued*

rendering of a wholemount, 4-month-old *Thy1-EGFP-M* mouse brain processed with EZ Clear and imaged by lightsheet fluorescence microscopy across a total imaging depth of 5 mm, dorsal to ventral axis and presented at (**D**) dorsal view and (**E and F**) sectioned digitally at the transverse (coronal) axis. (**G**) Brightfield representative images of adult mouse organs (eye, heart, lung, liver, spleen, kidney, and testis) processed by EZ Clear. (**H**) Maximum intensity projections of EZ Clear processed, wholemount lightsheet fluorescent microscopy imaged mouse organs (eye, heart, kidney, testis, and ovary) with the endothelium labeled by fluorescently conjugated *Lycopersicon esculentum* lectin (lectin-DyLight 649).

The online version of this article includes the following source data and figure supplement(s) for figure 1:

**Source data 1.** Quantitative comparison of the size changes of the mouse brains before and after processing with 3DISCO, Fast 3D, X-CLARITY, and EZ Clear.

**Figure supplement 1.** Comparison of EZ Clear to solvent- and aqueous-based clearing methods.

**Figure supplement 1—source data 1.** Comparison of sample size changes of EZ Clear to solvent- and aqueous-based clearing methods before, during, and after clearing.

brains were collected and then immersion fixed in 4% PFA overnight at 4°C. After fixation, labeled brains were treated with EZ Clear's lipid removal and washing steps, and then equilibrated and imaged in either RIMS or EZ View (*Figure 2E–J*). Imaging by LSFM demonstrated that not only were brains equilibrated in EZ View more transparent than those equilibrated in RIMS (*Figure 2C, D*), but they also featured deeper imaging depths (over 6 mm) through the dorsal-ventral axis of the brain compared to samples equilibrated in RIMS (*Figure 2F, I*). Color-coded depth projection stepping at 1 mm intervals showed that while the signal from lectin-649 starts to scatter 1–2 mm deep into RIMS treated tissue (*Figure 2G*), the signal remains strong and coherent (non-diffuse) at greater depths in the brain equilibrated and imaged in EZ View (*Figure 2J*). Quantitative comparison of the mean fluorescence intensity at depths of 1, 2, 3, 4, and 5 mm showed that while fluorescence intensity remained consistent in the EZ View equilibrated samples, mean fluorescence intensity was significantly lower after 4 mm in depth in the brains equilibrated and imaged in RIMS compared to those processed and imaged in EZ View (wholemount brains imaged for each condition n=3, two-way ANOVA and multiple comparisons). We also tested the long-term stability of fluorescence signal of samples stored in EZ View. Mouse brains perfused with lectin-649 were cleared with EZ Clear and stored in EZ View and then imaged 14, 17, 43, and 70 days later (*Figure 2L*). Notably, signal intensity showed no significant change through the imaging depth following prolonged storage in EZ View solution (n=3, two-way ANOVA and multiple comparisons). These results demonstrate that EZ View solution attains a high RI and effectively renders delipidated tissues optically transparent while minimizing light scattering while also maintaining fluorescent signal. Critically, EZ Clear samples remain in an aqueous environment throughout the entire clearing and imaging process, and no corrosive or combustible organic solvents, or potentially damaging immersion oils, are required for RI matching and imaging.

## Comparison between EZ Clear and Fast 3D

To further assess the clearing efficiency and sample transparency of EZ Clear, we compared samples cleared by two THF-based clearing methods, EZ Clear and Fast 3D, following administration of a synthetic, exogenous fluorescent dye that labels the endothelium. When Evans blue, a non-cell permeable dye, binds to albumin, it undergoes a conformational shift that produces fluorescence in the far-red spectrum (excitation at 620 nm, emission at 680 nm) (*Namykin et al., 2019*; *Saria and Lundberg, 1983*; *Honeycutt and O'Brien, 2021*). A cost affordable alternative to expensive dyes, such as fluorescent tomato lectin, Evans blue can thus be used to robustly label the endothelium following intravenous administration (*Robertson et al., 2015*). Adult mice perfused with Evans blue were euthanized and the brains were collected and fixed in 4% PFA overnight at 4°C. The brains were then bisected at the midline along the anterior-posterior axis, and the left hemisphere was cleared with Fast 3D, while the

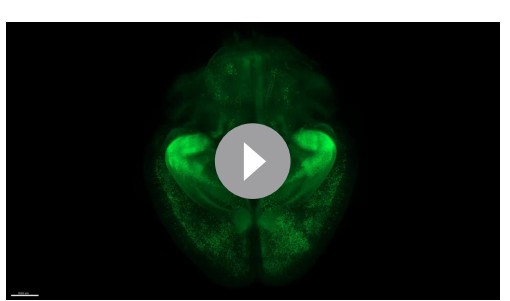

**Video 1.** Wholemount EZ Clear processed *Thy1-EGFP* adult mouse brain imaged by lightsheet fluorescent microscopy.

https://elifesciences.org/articles/77419/figures#video1

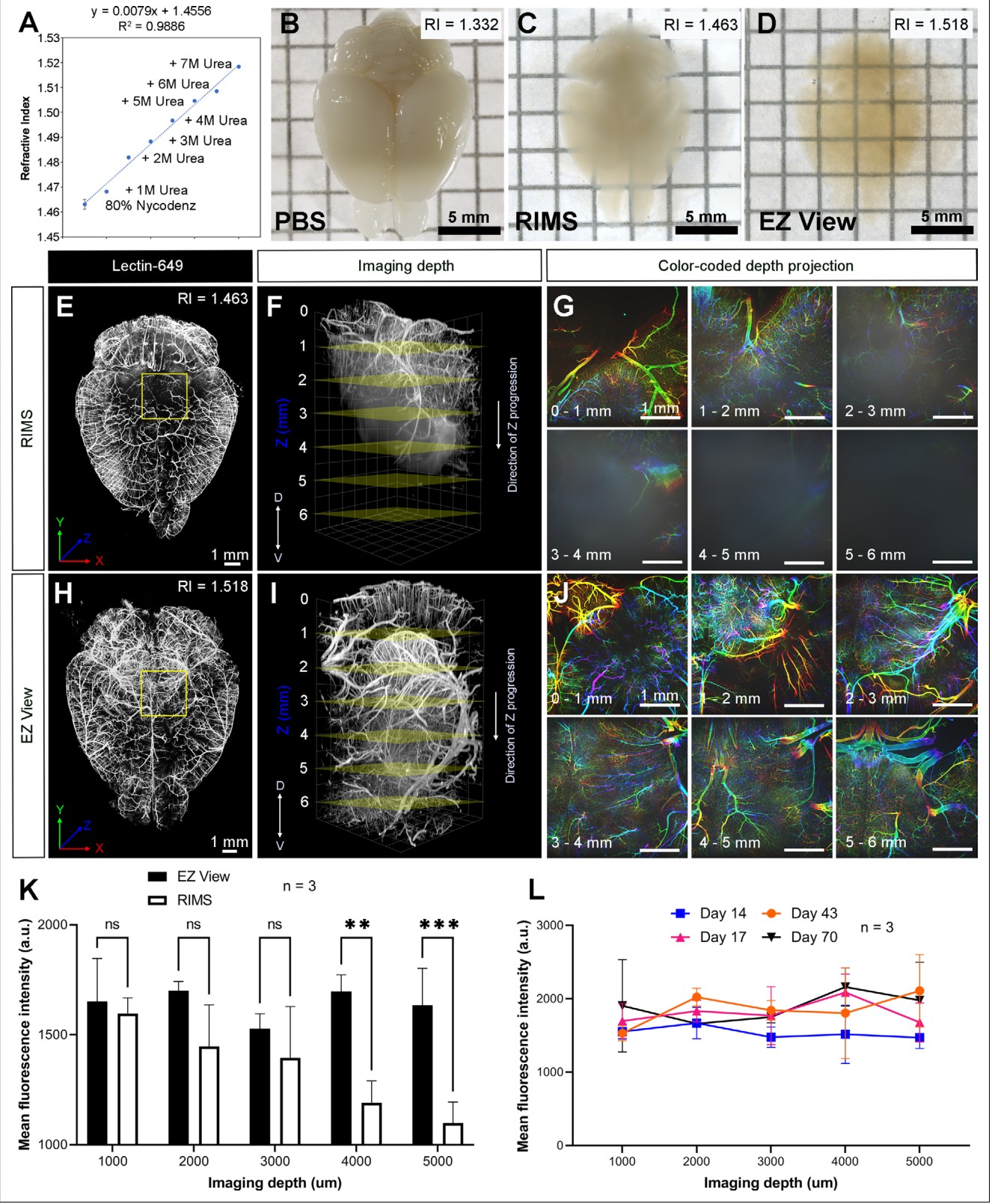

**Figure 2.** High refractive index imaging buffer EZ View achieves deeper imaging depth and maintains fluorescence stability. (**A**) Refractive index (RI) of 80% Nycodenz increases linearly with increasing concentrations of urea (n=3). (**B–D**) Comparison of transparency of adult mouse brains processed with EZ Clear lipid removal, washing, and then equilibration in (**B**) PBS (RI = 1.332), (**C**) refractive index matching solution (RIMS) (RI = 1.463), and (**D**) EZ View (RI = 1.518) for 24 hr at room temperature. Comparison of 3.5-month-old mouse brains perfused with lectin-649, then treated with delipidation,

*Figure 2 continued on next page*

*Figure 2 continued*

water washes, equilibrated in (**E–G**) RIMS and (**H–J**) EZ View, and imaged by lightsheet fluorescence microscopy (LSFM). (**E and H**) Comparison of volume rendered whole mouse brains, transverse view, and (**F and I**) across the imaging axis starting from dorsal to ventral, and (**G and J**) color-coded depth projection at 1 mm intervals beginning from dorsal (0 mm) to ventral (6 mm) side. (**K**) Quantitative comparison of mean fluorescence intensity of lectin-694 at different imaging depths (dorsal to ventral) shows the signal intensity gradually decreases along with the imaging depth when the brains were equilibrated in RIMS, unlike those equilibrated in EZ View. (n=3, error bars represent standard deviation [SD]. Two-way ANOVA and multiple comparisons. ns – not significant, **p<0.01, ***p<0.001.) (**I**) The fluorescence intensity of lectin-694 remain stable when stored in EZ View up to 70 days (n=3, two-way ANOVA and multiple comparisons).

The online version of this article includes the following source data for figure 2:

**Source data 1.** Measurements for refractive index changes in response to different concentrations of Nycodenz and urea, quantitative comparison of fluorescence intensity of lectin-649 at different imaging depths between brains equilibrated in EZ View and RIMS, and fluorescence intensity measurement of lectin-649 after storing samples in EZ View for 14, 17, 43, and 70 days.

---

right hemisphere was cleared with EZ Clear. Both hemispheres were imaged independently on a Zeiss Lightsheet Z1 from the dorsal to ventral side with an EC Plan-Neofluar 5x/0.16 air detection objective at a resolution of 1.829 μm laterally and 3.675 μm axially (*Figure 3A–F* and *Video 2*). The right hemisphere, processed with EZ clear, was equilibrated, and imaged in EZ View solution (RI = 1.518), while left hemisphere, cleared with Fast 3D, was equilibrated, and imaged in Fast 3D imaging solution (RI = 1.512). Although both EZ Clear and Fast 3D effectively rendered the tissue optical transparent (*Figure 1B*), lightsheet imaging demonstrated that EZ Clear also enabled imaging at depths up to 6.5 mm from dorsal to ventral sides, with focused signal and high contrast (*Figure 3A–C*), whereas light scattering increased at deeper imaging depths in the hemisphere processed using Fast 3D (*Figure 3D–F*). We further compared fluorescence intensity and contrast between the brains cleared with EZ Clear and Fast 3D. Mouse brains perfused with lectin-649 were either cleared with EZ Clear or Fast 3D (n=3 brains for each condition) and wholemount imaged through the dorsal-ventral axis by LSFM. A quantitative comparison of the mean fluorescence intensity at depths of 1, 2, 3, 4, or 5 mm into the brain (dorsal to ventral) showed no significant difference in lectin-649 signal intensity between EZ Clear and Fast 3D cleared samples (*Figure 3G*, n=3, two-way ANOVA and multiple comparisons). However, while the signal contrast remained high and consistent in EZ Clear processed brains, the lectin-649 signal contrast decreased with imaging depth in Fast 3D samples (*Figure 3H*, n=3). Lectin-649 signal contrast also remained high and consistent when quantitatively measured in other EZ Clear processed adult mouse organs, including the eye, heart, kidney, testis, and ovary (*Figure 3I*). Thus, in 48 hr and three simple steps, EZ Clear achieves high optical transparency, without sample shrinkage or swelling, it preserves endogenous and synthetic fluorescence, and it is compatible with aqueous-based imaging platforms. Additionally, EZ Clear enables imaging at deeper depths with minimal light scattering compared to other current tissue clearing modalities.

## EZ Clear treated samples are compatible with wholemount immunofluorescent staining

To determine the compatibility of EZ Clear with wholemount immunofluorescent staining, we tested the immunostaining protocol from iDISCO (*Renier et al., 2014*), as well as a standard immunofluorescent staining procedure, on samples that were delipidated and washed with EZ Clear (*Figure 4A*). We first stained adult mouse brains with To-Pro 3 to label nuclei and a Cy3-conjugated antibody against αSMA (αSMA-Cy3) to label vascular smooth muscle cell enwrapped blood vessels to determine how efficient the nuclear dye and fluorophore-conjugated antibody can penetrate through the brains after processing with EZ Clear. Both iDISCO and standard immunofluorescent staining protocols enabled effective diffusion and immunolabeling throughout EZ Clear treated samples (*Figure 4B–F*), as virtual sections from an LSFM imaged wholemount brain showed that

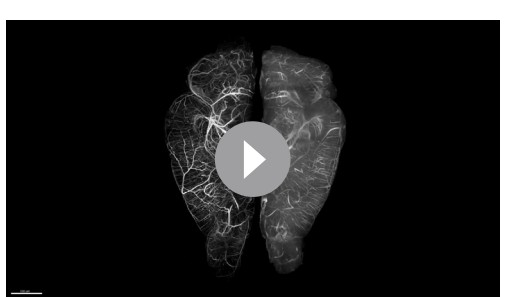

**Video 2.** Comparison of brain hemispheres following perfusion with Evans blue and clearing with either EZ Clear (right hemisphere) or Fast 3D (left hemisphere). https://elifesciences.org/articles/77419/figures#video2

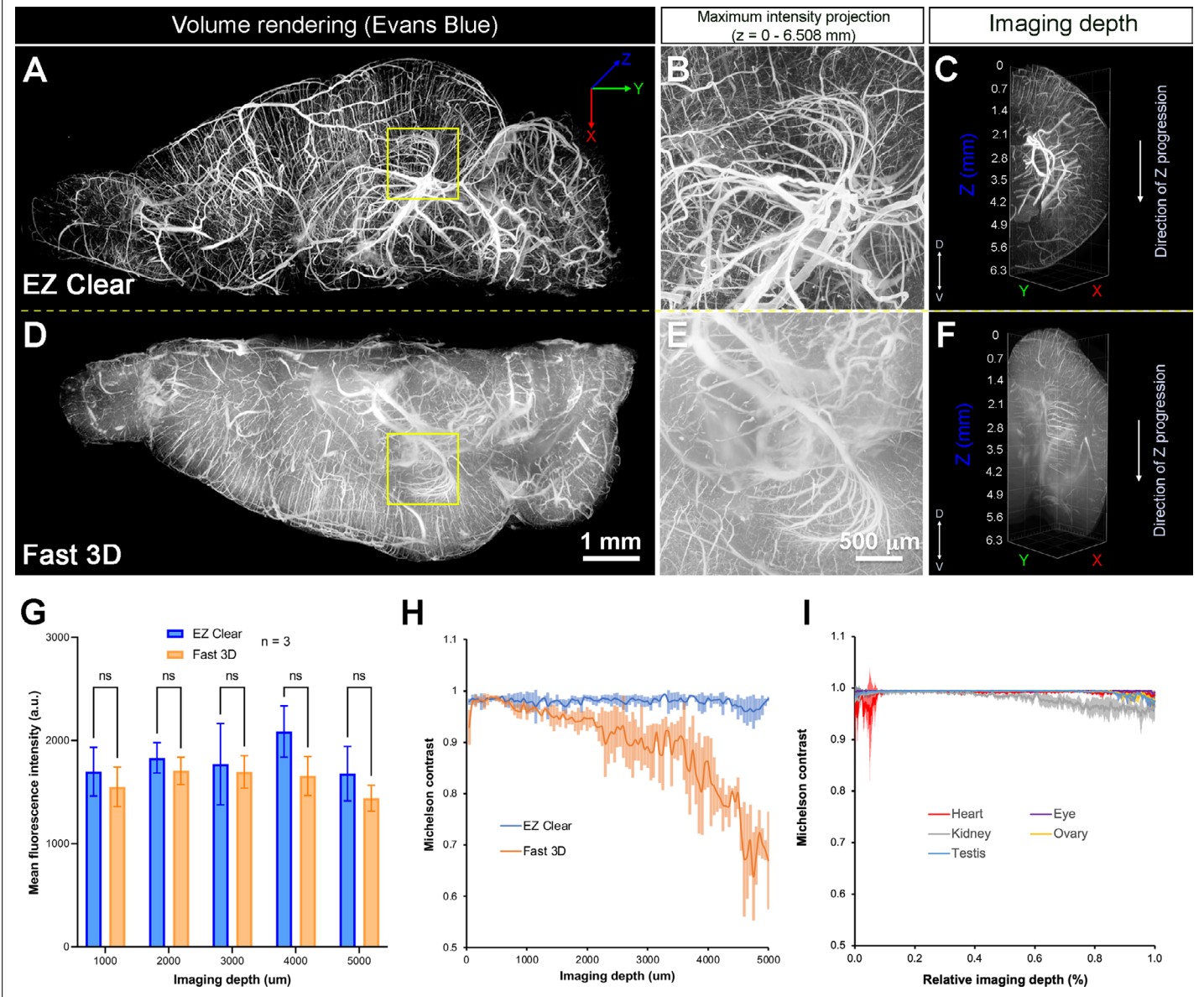

**Figure 3.** Quantitative comparison of fluorescence intensity and contrast between samples processed by EZ Clear and Fast 3D. (**A–F**) Lightsheet imaging of a mouse brain perfused with Evans blue dye and cleared by EZ Clear (right hemisphere, **A–C**) and Fast 3D (left hemisphere, **D–F**) shows that tissue processed with EZ Clear has cleaner signal and less light scattering. (**G**) Quantitative comparison of mean fluorescence intensity of lectin-649 at different imaging depths (dorsal to ventral axis) shows no significant difference between hemispheres treated with EZ Clear and Fast 3D (n=3, two-way ANOVA and multiple comparisons, error bars represent standard deviation [SD], ns – not significant). However, (**H**) while the lectin-649 fluorescence signal contrast remains sharp in EZ Clear treated brain hemispheres (n=3), the contrast of the signal gradually decreases along with imaging depth in the Fast 3D processed brains. (**I**) Contrast of the lectin-649 fluorescence signal also remains sharp in EZ Clear treated eye, heart, kidney, testis, and ovary.

The online version of this article includes the following source data for figure 3:

**Source data 1.** Measurements and comparisons of fluorescent intensity and contrast of lectin-649 between EZ Clear and Fast 3D cleared mouse brains and organs.

To-Pro 3 dye labeled nuclei evenly throughout the entire sample. Similarly, both protocols showed robust antibody penetrance as αSMA-Cy3 staining was evident throughout the wholemount samples (*Figure 4C*). Quantification of To-Pro 3 signal intensity across the brain sections (in *Figure 4B*, yellow dashed lines i and ii) also shows the intensity profiles are comparable between iDISCO and standard immunofluorescent protocols, with homogenous staining (*Figure 4D*). Quantitative comparison of the mean fluorescence intensity of To-Pro 3 (*Figure 4E*) and αSMA-Cy3 (*Figure 4F*) at the depths of

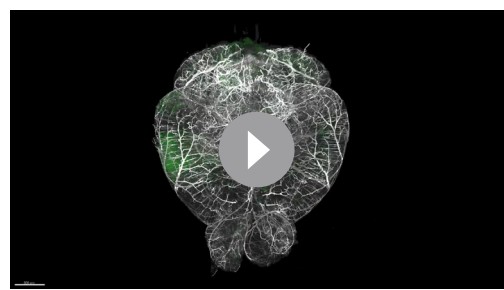

**Video 3.** Postnatal day 104 glioma brain perfused with DyLight 649-conjugated *Lycopersicon esculentum* (tomato) lectin (lectin-649).

https://elifesciences.org/articles/77419/figures#video3

1, 2, 3, 4, and 5 mm into the wholemount imaged brains (n=3, dorsal to ventral) also show no significant difference between samples stained with either protocol (*Figure 4E and F*, n=3, two-way ANOVA and multiple comparisons).

To further determine whether endogenous fluorescence can be preserved during the wholemount staining procedure, we processed brains from mice containing a *Thy1-EGFP* pan-neuronal reporter for wholemount immunostaining. After processing the samples with the EZ Clear lipid removal and washing steps, *Thy1-EGFP* brains were stained with a GFAP antibody to label astrocytes and To-Pro 3 to label nuclei using a standard immunofluorescent staining protocol. Lightsheet imaging showed that not only can the GFAP positive astrocytes be identified throughout the wholemount immunofluorescence-stained brain (*Figure 4G and H*, Cortex; *Figure 4I and J*, corpus callosum), robust signals from the *Thy1-EGFP* positive neurons can be detected throughout the sample. These results demonstrate that EZ Clear processed samples are compatible with wholemount immunofluorescent staining while retaining endogenous fluorescence.

## EZ Clear treated and imaged samples can be further processed for cryosectioning, histology, and immunofluorescent staining

To further determine the downstream uses of samples following EZ Clear treatment, we next tested whether the cleared and wholemount imaged samples could be processed for embedding and cryosectioning. For these studies, we examined brain tumor formation in a native murine model of glioma, as before (*Carlson et al., 2021*). To induce glioma, the lateral ventricle of E16.5 mouse embryos were injected with a DNA cocktail consisting of three plasmids: (1) a single pX330-variant construct encoding 3xFlag-NLS-Cas9-NLS, along with three human *RNA Polymerase III U6 snRNA* (*RNU6-1*) promoter cassettes upstream of guide RNAs targeting the tumor suppressor genes *Nf1*, *Trp53*, and *Pten*, (2) a *piggyBac* (PB) helper plasmid with the radial glial- and astrocyte-specific promoter, *Solute Carrier Family1 Member 3*, *SLC1A3* (also known as EAAT1, GLAST, Genbank AF448436.1), driving expression of PB transposase, and (3) a PB cargo fluorescent EGFP reporter vector to indelibly label all tumor cells and their descendants (*Carlson et al., 2021*). Following injection of the DNA cocktail, the embryos were electroporated to allow uptake of the constructs, then the embryos were placed back in the maternal cavity. These in utero electroporated (IUE) animals with tumor suppressor deficient cells were then birthed normally and were collected at postnatal day 104 (P104) for perfusion with lectin-649. Next, the brains were collected, fixed, and processed with EZ Clear and wholemount imaged by LSFM to reveal the distribution of GFP+ tumor cells and the lectin-649 labeled vasculature (*Figure 5A and B*). At a macro level, whole brain imaging showed that GFP+ tumor cells can be identified not only as clusters (*Figure 5A and B*), but also as single cells (*Figure 5C and D*, *Video 3*). After imaging, the glioma containing samples were immersed in ×1 PBS overnight to remove the EZ View solution, then sucrose protected and embedded for cryosectioning. P104 glioma brains processed with EZ Clear were then cryosectioned and sections were used for direct immunofluorescence or for hematoxylin and eosin (H&E) staining to visualize tissue architecture. After cryosectioning, tissue sections were mounted and inspected to evaluate whether

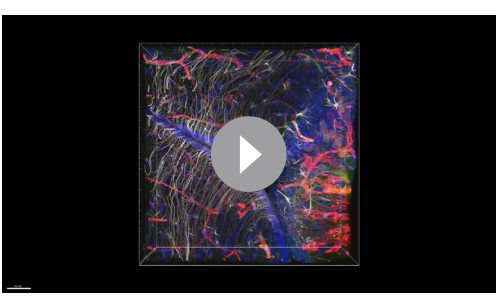

**Video 4.** Three-dimensional (3D) volume rendering of a confocal stack from a glioblastoma multiforme (GBM) brain following cryosectioning and immunostained with GFAP (white) and Hoechst (blue) after EZ Clearing and lightsheet fluorescence microscopy (LFSM) imaging of the tumor (green) and blood vessels (red).

https://elifesciences.org/articles/77419/figures#video4

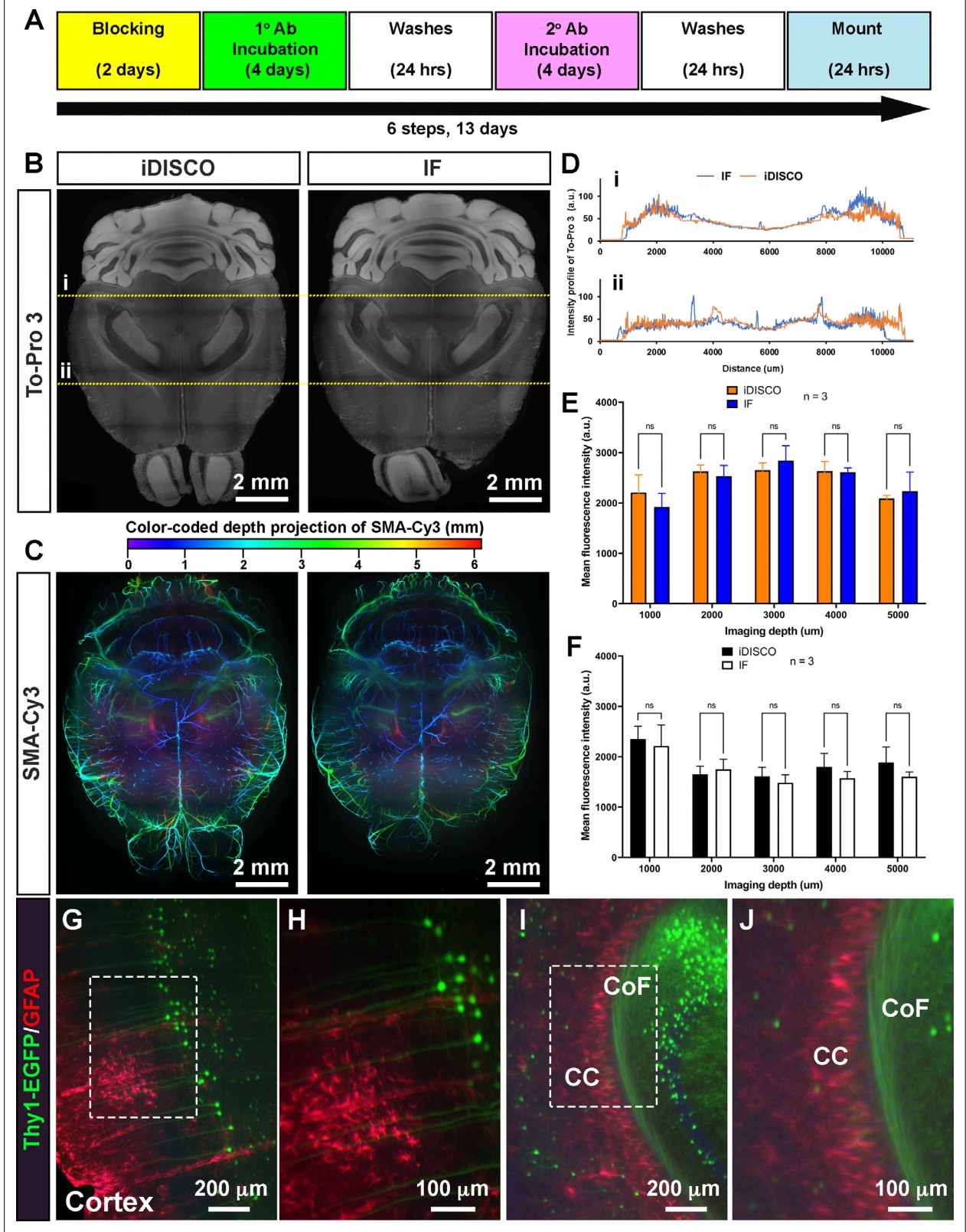

**Figure 4.** EZ Clear processed samples are compatible with wholemount immunofluorescent staining. (**A**) Six-step, standard wholemount immunofluorescent staining procedure for EZ Clear treated mouse brains. (**B**) Whole organ immunostaining of EZ Clear processed mouse brains using iDISCO or standard immunostaining (direct IHC) protocols along with To-Pro 3 staining to label nuclei (digital sectioned transversely at 2.5 mm) and (**C**) anti-smooth muscle α-actin conjugated to Cy3 to label smooth muscle cells and arteries (color-coded depth projection). (**D**) Comparison of the

*Figure 4 continued on next page*

*Figure 4 continued*

To-Pro 3 penetrance across the lateral axis between iDISCO and standard IHC processed samples. Quantitative comparison of mean fluorescence intensity of (**E**) To-Pro 3 and (**F**) αSMA-Cy3 at different imaging depths (dorsal to ventral) shows no significant difference between EZ Clear treated brains stained with iDISCO and standard IHC protocols (n=3, two-way ANOVA and multiple comparisons, error bars represent standard deviation [SD], ns = not significant.). (**G–J**) *Thy1-EGFP-M* mouse brain processed with EZ Clear, wholemount immunostained using a standard immunofluorescent protocol with antibodies raised against GFAP (red, glia) and wholemount imaged by lightsheet fluorescence microscopy (LSFM) at the (**G and H**) cortex and the intersection of the (**I and J**) corpus callosum (CC) and the commissure of the fornix (CoF).

The online version of this article includes the following source data for figure 4:

**Source data 1.** Measurements To-Pro 3 penetrance across the lateral axis between iDISCO and standard IHC processed samples.

the fluorescence from the transgenic *piggyBac* EGFP reporter and synthetic fluorescent lectin dye were preserved. Sections mounted in EZ View solution were highly transparent compared to those which remained in PBS or were mounted in Prolong Glass Antifade medium (*Figure 5E–G*). Sections mounted in EZ View were compatible with both fluorescence (*Figure 5H*) and confocal imaging (*Figure 5I–J*), as signals from EGFP⁺ tumor cells and perfused lectin-649 vessels were well preserved. EZ Clear processed, wholemount imaged, and cryosectioned tissues were also compatible with conventional histology, as evidenced by robust H&E labeling of tissues (*Figure 5K–M*) comparable to tissues fixed using only 4% PFA (*Figure 5—figure supplement 1*).

Next, we explored whether EZ Clear processed and cryosectioned tissues are compatible with indirect immunofluorescence. Free-floating sections from brains harboring EGFP⁺ glioma cells that were perfused with lectin-649 were processed through the standard immunofluorescent staining protocol (*Figure 6A*). Sections were stained with antibodies against CD31 (to label endothelial cells), GFAP (astrocytes), smooth muscle α-actin conjugated with Cy3 (αSMA-Cy3) and class III β-tubulin (e.g., Tuj1, to label neurons), and then mounted with EZ View solution. Confocal imaging revealed that in addition to preserving the fluorescence of EGFP⁺ tumor cells and perfused lectin-649 labeled endothelial cells, each of these distinct antigens (GFAP, αSMA, and Tuj1) was robustly detected by indirect immunofluorescence following EZ Clearing and cryosectioning (*Figure 6B-D*, *Video 4*). These results demonstrate that EZ Clear processed tissue not only can be interrogated at a macro level by whole organ 3D imaging using LSFM, but can be further used for cryosectioning and investigation at the cellular level via either histological or immunofluorescent staining.

## Discussion

We have developed a simple, rapid, and robust tissue clearing procedure, EZ Clear, that renders entire adult mouse organs transparent within 48 hr, without the need for special equipment or toxic organic solvents. By combining a water-miscible solvent that rapidly infiltrates and dissolves lipids within a tissue, and an aqueous high RI sample mounting medium that minimizes light scattering and renders tissues optically transparent, EZ Clear makes whole organ clearing and high-resolution imaging simple and compatible with a variety of imaging needs.

EZ Clear has several advantages over other clearing methods. First, unlike electrophoretic-based CLARITY clearing, EZ Clear does not require a significant initial financial investment in specialized equipment. Indeed, its simplicity and robustness make it easy to adopt and apply immediately in any standard molecular biological laboratory. Additionally, samples require minimal attention during processing, and this simple, three-step protocol requires little time and effort, as it yields cleared tissue within 48 hr. THF has been used in other clearing methods, such as 3DISCO and Fast 3D, for tissue dehydration and lipid removal from samples prior to RI matching to render tissue transparent. THF is highly water-miscible, easily infiltrates biological samples, solubilizes lipids, and minimizes fluorescent quenching (*Ertürk et al., 2012*; *Díaz et al., 1992*; *Qi et al., 2019*). However, instead of going through graded series of THF for lipid removal, which dehydrates tissue and can alter sample size (*Ertürk et al., 2012*; *Kosmidis et al., 2021*), we determined that 50% THF prepared in ddH₂O is not only sufficient for dissolving lipids within tissues, but samples remain hydrated during the entire procedure, which minimizes size changes. This simple, one-step immersion procedure also significantly reduces the amount of THF required for processing each sample. Furthermore, while the RI of water is 1.33, the RI of the soft tissue is estimated to be between 1.44 and 1.56, and the RI of dry tissue is approximately 1.50 (*Ueda et al., 2020*; *Jacques, 2013*; *Silvestri et al., 2016*). In

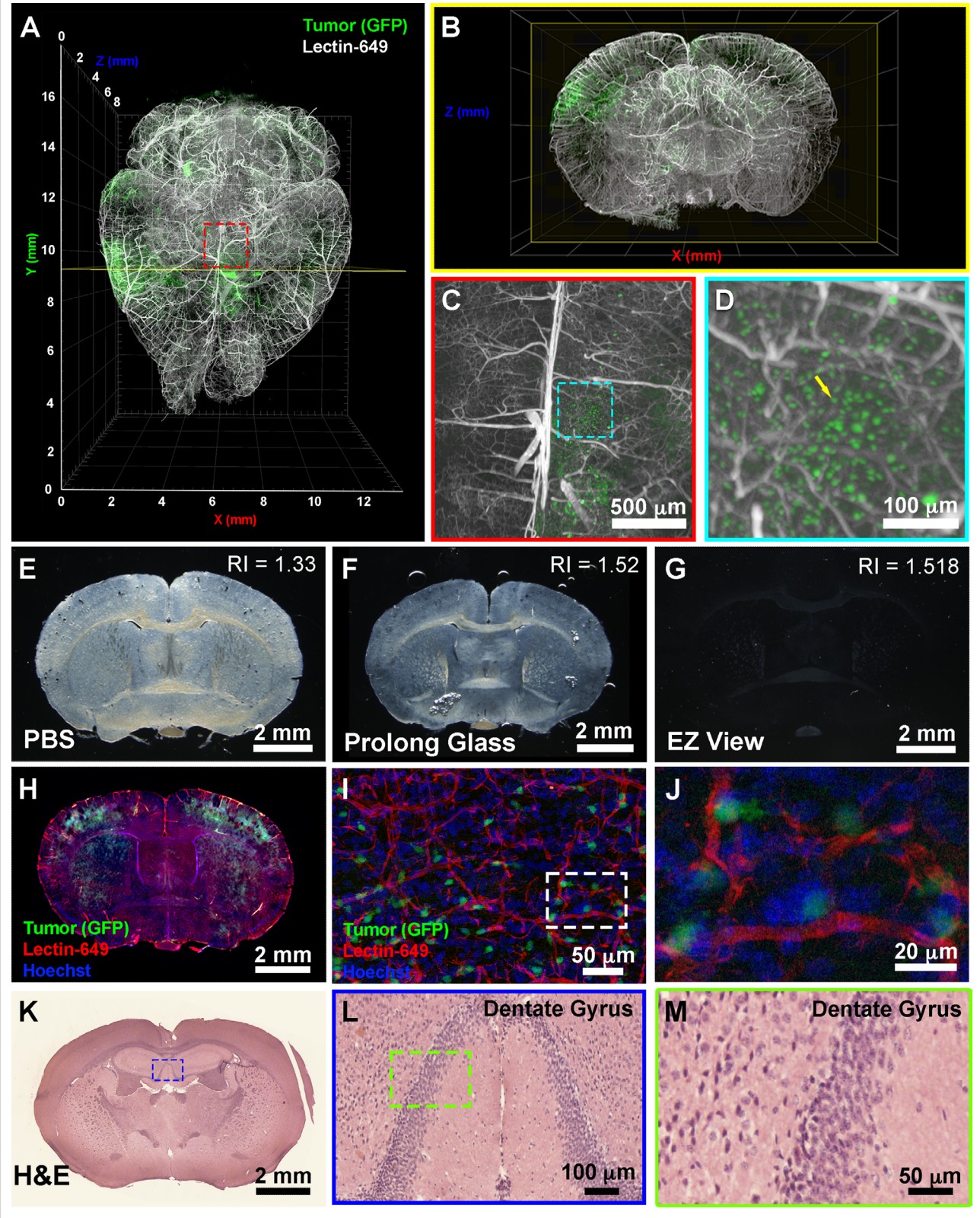

**Figure 5.** EZ Clear processed and imaged samples can be further processed for downstream cryosection, histology, and immunofluorescence staining. (**A**) Volume rendering of a wholemount lightsheet fluorescence microscopy (LSFM) imaged, EZ Clear processed postnatal day 104 (P104) glioblastoma multiforme (GBM) mouse brain with GFP⁺ tumor cells and lectin-649 labeled vasculature at (**A**) dorsal view and (**B**) sectioned digitally in the transverse (coronal) axis. GFP⁺ tumor cells can be identified in large clusters (**A and B**), as well as sparse single cells (**C and D**) from wholemount imaged data.

*Figure 5 continued on next page*

*Figure 5 continued*

(**E–G**) EZ Clear processed and imaged brains processed for cryosectioning in the coronal plane, and then mounted in (**E**) PBS, (**F**) Prolong Glass Antifade medium (n=1.52), or (**G**) EZ View. Sections mounted in EZ View are compatible with fluorescence (**H**) and confocal (**I and J**) imaging and the signals from GFP⁺ tumor cells and fluorescent lectin labeled vasculature are preserved. (**K–M**) Tissues processed for EZ Clear are also compatible with downstream histological applications, as cryosections stained with hematoxylin and eosin (H&E) yielded robust nuclear and cytoplasmic labeling.

The online version of this article includes the following figure supplement(s) for figure 5:

**Figure supplement 1.** EZ Clear processed and imaged samples processed for cryosectioning and hematoxylin and eosin (H&E) staining.

solvent-based clearing methods (e.g., 3DISCO and BABB), dehydrated and delipidated samples are equilibrated in high RI solvents, such as DBE (RI = 1.56) and BABB (RI = 1.55) to reduce light scattering. In aqueous-based clearing methods, solutions like RIMS (80% (v/v) Nycodenz, RI = 1.46) and sRIMS (80% D-sorbitol, RI = 1.43) are used for mounting and imaging of SDS-mediated delipidated tissue, although they have less than ideal RIs, falling well below the ideal range of 1.52–1.56 (*Yang et al., 2014*; *McCreedy et al., 2021*). By using an aqueous sample mounting and imaging solution with a high RI (1.518), delipidated samples remain hydrated while being imaged, preserving endogenous fluorescent signals, and making EZ Clear compatible with most fluorescent imaging platforms. Imaging samples in EZ View obviates the use of immersion oil (*Kosmidis et al., 2021*), further simplifying additional downstream processing and imaging. We also note that EZ View solution can be used as a sample mounting and imaging medium for tissue after cryosectioning and attachment to slides. Finally, we demonstrated that not only are EZ Clear processed samples compatible with wholemount immunofluorescent staining with both passive standard immunofluorescent and iDISCO staining protocols, but they can also be processed for cryosectioning and standard histological staining or immunofluorescent staining, effectively reducing the number of samples required for imaging studies. While whole organ immunostaining and imaging at the macro level is possible with currently established protocols (*Renier et al., 2014*; *Susaki, 1982*), we propose that initial whole organ imaging with EZ Clear followed by interrogation at the cellular level in thicker sections via cryosectioning and immunohistochemistry-based indirect immunofluorescence allows researchers to maximize the yield of precious tissue samples while minimizing the costs associated with excessive antibody use.

A shortcoming of the present study is that we have not yet tested EZ Clear on larger samples, or on tissues from other species such as rats, pigs, or humans, nor did we examine mouse tissues from younger ages (e.g., embryos). Future testing on the volume of the lipid removal solution required, and incubation time necessary, scaled to various sample sizes with varying lipid contents will be necessary for further optimization of the technique, but we anticipate that this method will readily work with samples smaller than the adult mouse organs used in the present study.

In summary, EZ Clear renders adult mouse organs optically transparent in 48 hr in just three simple steps using standard, off the shelf reagents in an aqueous-based tissue clearing methodology. EZ Clear is a simple, robust, and easy to adopt whole organ clearing technique that preserves endogenous fluorescent reporters and is compatible with most common imaging platforms, while offering the additional benefit of preserving samples for further downstream imaging analyses.

## Materials and methods

**Key resources table**

| Reagent type (species) or resource | Designation | Source or reference | Identifiers | Additional information |
|---|---|---|---|---|
| Antibody | Lycopersicon esculentum (Tomato) Lectin (LEL, TL), DyLight 649 | Vector Laboratories | Cat #: DL-1178–1 | |
| Antibody | Rat Monoclonal Anti-Mouse CD31 | BD Biosciences | Cat #: 550274 | IF (1:200) |
| Antibody | Mouse Monoclonal Anti-actin, alpha-smooth muscle – Cy3 | Millipore-Sigma | Cat #: C6198 | IF (1:200) |
| Antibody | Rat Monoclonal Anti-GFAP | Invitrogen | Cat #: 13-0300 | IF (1:200) |
| Antibody | Rabbit Polyclonal Anti-beta III Tubulin | Abcam | Cat #: ab18207 | IF (1:200) |

*Continued on next page*

*Continued*

| Reagent type (species) or resource | Designation | Source or reference | Identifiers | Additional information |
|---|---|---|---|---|
| Antibody | Donkey Anti-Rat Alexa Fluor 568 | Abcam | Cat #: Ab175475 | IF (1:500) |
| Antibody | Donkey Anti-Rabbit Alexa Fluor 488 | Invitrogen | Cat #: A21206 | IF (1:500) |
| Antibody | Donkey Anti-Rabbit Alexa Fluor 568 | Invitrogen | Cat #: A10042 | IF (1:500) |
| Chemical compound, drug | Evans blue | Millipore-Sigma | Cat#: E2119 | |
| Chemical compound, drug | Histodenz | Millipore-Sigma | Cat#: D2158-100g | |
| Chemical compound, drug | Hoechst 33342 | Millipore-Sigma | Cat#: 14533 | 10 mg/mL stock at 1:1000 dilution |
| Chemical compound, drug | Diatrizoic acid | Millipore-Sigma | Cat#: D9268-1g | |
| Chemical compound, drug | Benzyl ether | Millipore-Sigma | Cat#: 108014 | |
| Chemical compound, drug | $N$-methyl-D-glucamine | Millipore-Sigma | Cat#: M2004-100g | |
| Chemical compound, drug | Triethylamine | Millipore-Sigma | Cat#: T0886 | |
| Chemical compound, drug | Tetrahydrofuran | Millipore-Sigma | Cat#: 186562 | |
| Chemical compound, drug | Nycodenz | Accurate Chemical & Scientific | Cat#: 100334–594 | |
| Chemical compound, drug | D-Sorbitol | Millipore-Sigma | Cat#: S1876 | |
| Chemical compound, drug | Urea | Millipore-Sigma | Cat#: U5378 | |
| Chemical compound, drug | Sodium azide | Millipore-Sigma | Cat#: S2002 | |
| Chemical compound, drug | VWR Life Science Agarose I | VWR | Cat#: 0710 | |
| Chemical compound, drug | Triton X-100 | Thermo Fisher Scientific | Cat#: BP-151 | |
| Chemical compound, drug | Donkey serum | Millipore-Sigma | Cat#: D9663 | |
| Chemical compound, drug | $Na_2HPO_4$ | Acros Organics | Cat#: 448140010 | |
| Chemical compound, drug | $NaH_2PO_4$ | Millipore-Sigma | Cat#: S0751 | |
| Chemical compound, drug | PBS, ×1 solution, pH 7.4, | Thermo Fisher Scientific | Cat#: BP24384 | |
| Chemical compound, drug | Paraformaldehyde | Millipore-Sigma | Cat#: P6148 | |
| Chemical compound, drug | Modified Harris Hematoxylin Solution | Millipore-Sigma | Cat#: HHS32-1L | |
| Chemical compound, drug | Eosin Y Phloxine B Solution | EMS | Cat#: 26051–21 | |
| Chemical compound, drug | Histoclear II | EMS | Cat#: 64111–04 | |

*Continued on next page*

*Continued*

| Reagent type (species) or resource | Designation | Source or reference | Identifiers | Additional information |
|---|---|---|---|---|
| Chemical compound, drug | DPX New | Millipore-Sigma | Cat#: 100579 | |
| Chemical compound, drug | Tissue-Tek OCT Compound | Sakura | Cat#: 4583 | |
| Software, algorithm | Imaris | Bitplane Inc | RRID:SCR_007370 | Ver. 8.5 |
| Software, algorithm | Vision4D | Arivis | RRID:SCR_018000 | Ver. 2.12 and 3.5 |
| Software, algorithm | Zeiss Zen Blue | Zeiss | RRID:SCR_013672 | Ver. 2.6.76.00000 (LSM 880) Ver. 1.1.2.0 (AxioZoom.V16 and AxioObserver.Z1) |
| Software, algorithm | Zeiss Zen Black | Zeiss | RRID:SCR_018163 | Ver. 9.2.5.54 (Lightsheet Z.1) Ver. 14.0.23.201 (LSM 880) |
| Software, algorithm | Labscope Material | Zeiss | | Ver. 2.8.4 |
| Software, algorithm | ImageJ/Fiji | NIH | RRID:SCR_002285 | Ver. 1.53c |
| Software, algorithm | Prism | GraphPad | RRID:SCR_002798 | Ver. 9.2.0 |
| Other | TO-PRO–3 Iodide (642/661) | Thermo Fisher Scientific | Cat#: T3605 | IF (1:1000) Nuclear counterstain |
| Other | Plastic Binding Head Slotted Screws, Off-White, 4–40 Thread Size, 1" Long | McMaster-Carr | Cat#: 94690A724 | Sample holder parts for lightsheet imaging |
| Other | Nylon Plastic Washer for Number 4 Screw Size, 0.112" ID, 0.206" OD | McMaster-Carr | Cat#: 90295A340 | Sample holder parts for lightsheet imaging |
| Other | Nylon Hex Nut, 4–40 Thread Size | McMaster-Carr | Cat#: 94812 200 | Sample holder parts for lightsheet imaging |

## Mice

This study was carried out in strict accordance with the recommendations in the Guide for the Care and Use of Laboratory Animals of the National Institutes of Health. All animal research was conducted according to protocols approved by the Institutional Animal Care and Use Committee (IACUC) of Baylor College of Medicine.

## Generation of endogenous glioma in a mouse model using IUE

All mouse CRISPR-IUE GBM gliomas were generated in the CD-1 IGS mouse background. IUEs were performed as previously described (*Carlson et al., 2021*). Briefly, a plasmid containing guide RNAs targeting the tumor suppressor genes *Nf1*, *Tp53*, and *Pten* was co-electroporated along with a fluorescent reporter EGFP to label tumor cells. The uterine horns were surgically exposed in a pregnant dam at E16.5 and the embryos were injected with a DNA cocktail containing the following four plasmids: (1) a single pX330-variant (*Cong et al., 2013*) construct encoding 3xFlag-NLS-Cas9-NLS, along with three human *RNA Polymerase III U6 snRNA* (*RNU6-1*) promoter cassettes upstream of validated guide RNA sequences targeting *Nf1* (GCAGATGAGCCGCCACATCGA), *Trp53* (CCTCGAGCTCCCTCTGAGCC), and *Pten* (GAGATCGTTAGCAGAAACAAA) (*Xue et al., 2014*); (2) a *piggyBac* (PB) helper plasmid with the glial- and astrocyte-specific promoter, *Solute Carrier Family1 Member 3*, *SLC1A3* (also known as EAAT1, GLAST, Genbank AF448436.1), driving expression of PB transposase (pGlast-PBase) (*Chen et al., 2015*), and a PB cargo fluorescent reporter vector (pbCAG-GFP-T2A-GFP or pbCAG-mRFP1). The PBase helper plasmid promotes stable integration of the cargo fluorescent reporter vector, which indelibly labels all descendant cells, allowing one to visualize tumors over time via fluorescence. Following injection of the glioma-inducing CRISPR-Cas9/PB cocktail (2.0 µg/µL pGLAST-PBase, 1.0 µg/µL all other plasmids) into the lateral ventricle of each embryo, embryos were electroporated six times at 100 ms intervals using BTW Tweezertrodes connected to a pulse generator (BTX 8300) set at 33 V and 55 ms per pulse. Voltage was applied across the entire brain to allow uptake of the constructs. The uterine horns were placed back in the cavity, and these dams developed normally, but their electroporated offspring featured malignancies postnatally, as the tumor

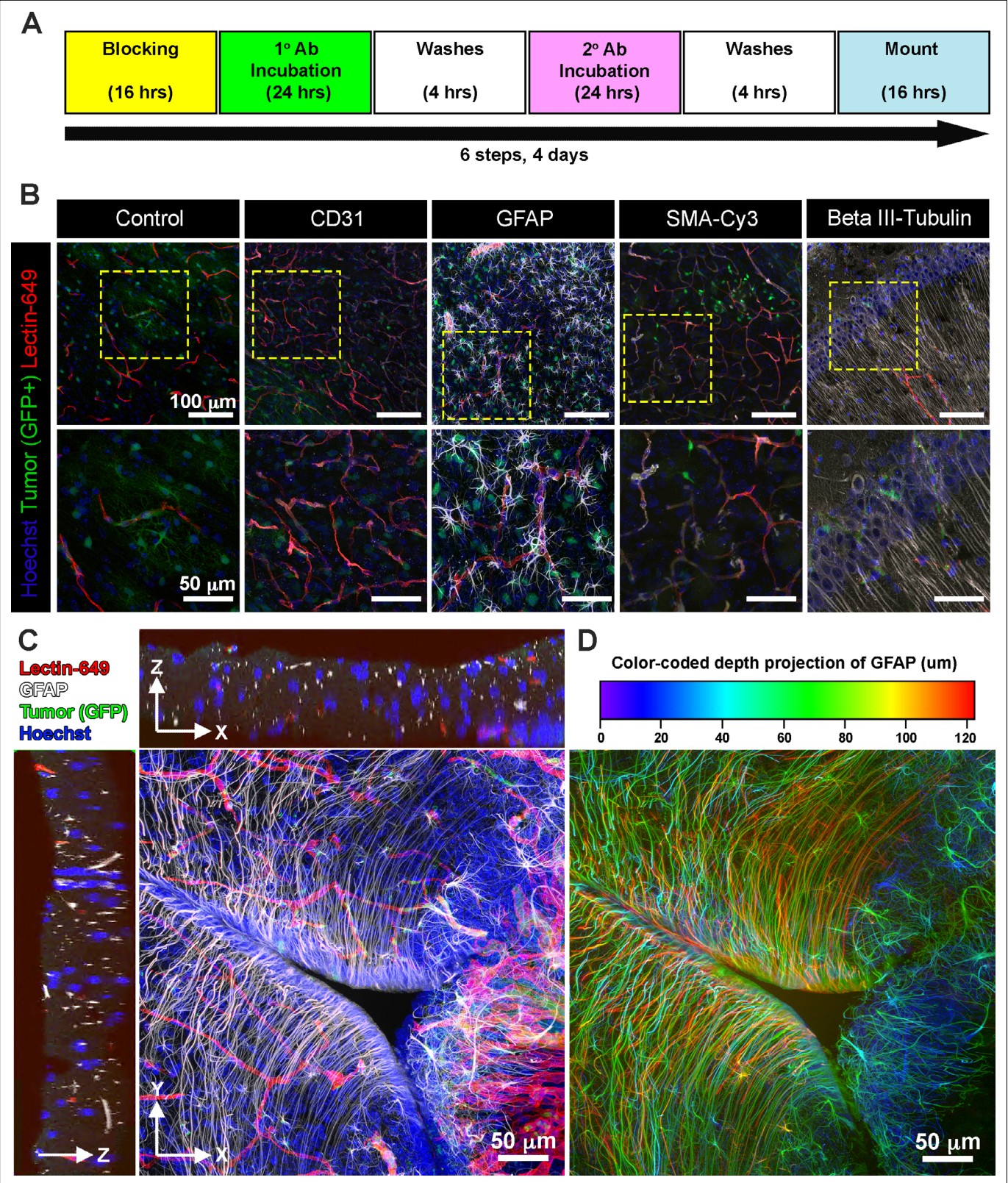

**Figure 6.** EZ Clear processed and imaged samples are compatible with cryosectioning and immunofluorescent staining. (**A**) Six-step immunostaining procedure for processing tissues after EZ Clearing and imaging. (**B**) Cryosections in the coronal plane from an EZ Clear processed and imaged postnatal day 104 (P104) glioblastoma multiforme (GBM) mouse brain with GFP+ tumor cells and lectin-649 labeled vessels were directly imaged (control) or immunostained to detect CD31 (endothelial), GFAP (astrocytes, glia), smooth muscle α-actin (smooth muscle cells), and β-III tubulin (neurons). Stained

*Figure 6 continued on next page*

*Figure 6 continued*

sections were mounted on slides with EZ View and imaged on an 880 Airyscan FAST confocal microscope at ×20. (**C**) Orthogonal and (**D**) color-coded depth projection views of GFP⁺ tumor cells and lectin-649 labeled vessels section labeled with GFAP (astrocytes, glia) and Hoechst (nuclei). The GFAP signal was constant throughout the 100 μm section.

suppressor deficient cells expanded. Animals were perfused at P104 with *L. esculentum* (tomato) lectin fluorescently conjugated with DyLight 649 (Vector Laboratories, Cat. No. DL-1178) before euthanized. A detailed procedure for animal perfusion and lectin labeling is listed below. The brains were then dissected out for further processing.

## Animal perfusion

To perfuse the adult mice, they were deeply anesthetized by $CO_2$ inhalation and the chest cavity was opened to expose the beating heart. The right atrium was incised, and the mouse was transcardially perfused through the left ventricle with 10 mL of room temperature ×1 PBS, followed by 10 mL of cold 4% PFA. Organs were then dissected from the mouse and then drop fixed in 4% PFA at 4°C for 24 hr with gentle agitation. After fixation, the mouse organs were washed in ×1 PBS three times, 30 min each at room temperature, then stored in ×1 PBS with 0.05% sodium azide at 4°C before proceeding with clearing procedure.

For animals that were labeled with lectin, adult mice were injected in their tail vein with 100 μL of *L. esculentum* (tomato) lectin fluorescently conjugated with DyLight 649 (lectin-649, Vector Laboratories, Cat. No. DL-1178). Lectin was allowed to circulate for a minimum of 5 min to enable adequate circulation and binding of the lectin to the endothelium. Mice were then deeply anesthetized by $CO_2$ inhalation and the chest cavity was opened to expose the beating heart. An additional 75 μL of lectin-649 was injected through the left ventricle using a 31-gauge insulin syringe (BD Biosciences, Cat. No. 324911). Lectin was perfused by hand slowly over 1 min and the needle was kept in place for an additional minute after injection to allow the pumping heart to circulate the dye. Afterward, the right atrium was incised, and the animal was transcardially perfused through the left ventricle with an additional 10 mL of room temperature ×1 PBS, followed by 10 mL of cold 4% PFA. Organs were then dissected and drop fixed in 4% PFA at 4°C for 24 hr with gentle agitation. After fixation, all organs were washed in ×1 PBS three times, 30 min each at room temperature, then stored in ×1 PBS with 0.05% sodium azide at 4°C before proceeding with clearing.

For animals that were labeled with Evans blue, adult mice were injected in their tail vein with 100 μL of Evans blue solution (2% (w/v) with 0.9% (w/v) NaCl in sterile Milli-Q water, filter sterilized with a 0.22 μm filter), as in *Honeycutt and O'Brien, 2021*. The dye was allowed to circulate for a minimum of 5 min. Mice were then deeply anesthetized by $CO_2$ inhalation and the chest cavity was opened to expose the beating heart. An additional 100 μL of Evans blue solution was injected through the left ventricle using a 31-gauge insulin syringe (BD Biosciences, Cat. No. 324911). The dye was perfused by hand slowly over 1 min and the needle was kept in place for an additional minute after injection to prevent dye leaking back out. The injected Evans blue solution was allowed to circulate until the hindlimbs and tail turned blue, or 5 min was reached. Animals were then euthanized with $CO_2$ inhalation followed by cervical dislocation. The brain was then dissected from the skull and drop fixed in 4% PFA at 4°C for 24 hr with gentle agitation. After fixation, the brain was washed in ×1 PBS three times, 30 min each at room temperature, then stored in ×1 PBS with 0.05% sodium azide at 4°C before proceeding with clearing procedure.

## EZ Clear protocol

After perfusion and overnight fixation, samples were placed in individual glass scintillation vials, protected from light with foil, and rocked on an orbital shaker in a vented chemical fume hood, and incubated with 20 mL of the following solutions in sequence for the EZ Clear protocol: (1) Lipid removal: 50% (v/v) THF (with 250 ppm BHT, Millipore-Sigma, 186562) prepared in sterile Milli-Q $H_2O$ for 16 hr. (2) Wash: rinsed with sterile Milli-Q $H_2O$ four times, 1 hr each at room temperature. (3) RI matching: incubated with 5 mL of EZ View sample mounting and imaging solution for 24 hr to render the samples transparent for imaging. The EZ View solution consisted of 80% Nycodenz (Accurate Chemical & Scientific 100334-594), 7 M urea, 0.05% sodium azide prepared in 0.02 M sodium phosphate buffer. To prepare the solution, 52.5 g of urea and 31.25 mg of sodium azide were mixed with

35 mL of 0.02 M sodium phosphate buffer (pH 7.4) in a 250 mL beaker. The solution was stirred and gently heated to 37°C on a hot plate until the urea completely dissolved. Then, 100 g of Nycodenz was slowly stirred in until completely dissolved. The final volume of the solution was adjusted to 125 mL with 0.02 M sodium phosphate buffer. The dissolved solution was filtered through a vacuum filtration system (Nalgene vacuum filtration system filter, pore size 0.2 μm, Z370606) and stored at room temperature. The RI of the EZ View solution was measured on a refractometer (Atago, PAL-RI 3850) and the RI was between 1.512 and 1.518. Cleared samples were protected from light, placed on a shaker, and rocked gently at room temperature for 24 hr to render them transparent. All of the adult mouse organs (brain, eye, heart, lung, liver, kidney, spleen, testis, and ovary) were processed following the same procedure.

### 3DISCO clearing

3DISCO clearing was performed according to Ertürk et al., 2012. Briefly, after perfusion and fixation, adult mouse brains were incubated in graded series of 20 mL of THF solutions mixed with filtered Milli-Q $H_2O$ at increasing concentrations of 50, 70, 80, and 100% (v/v), 1 hr each, in a scintillation vial at room temperature. The scintillation vial was covered with foil and placed on an orbital shaker within a vented chemical fume hood. Brains were then immersed in fresh 100% THF overnight, followed by a 1 hr incubation in fresh 100% THF. After removing THF completely, samples were then immersed in benzyl ether (DBE, Millipore-Sigma, 108014) overnight to render them transparent.

### X-CLARITY clearing

X-CLARITY clearing was performed according to the manufacturer's instruction (Logos Biosystems). Briefly, after perfusion and fixation, adult mouse brains were immersed in 5 mL of X-CLARITY Hydrogel Solution with 0.25% (w/v) of polymerization initiator VA-044 (Logos Biosystems, C1310X) at 4°C for 24 hr. The hydrogel infused brain was then thermo-induced crosslinked for 3 hr at 37°C under vacuum (–90 kPa). After the crosslinking reaction, the hydrogel solution was removed and the brain was washed in ×1 PBS three times, 1 hr each, then once overnight at 4°C. The hydrogel infused and crosslinked brain was then cleared in electrophoretic tissue clearing solution (Logos Biosystems, C13001) using the X-CLARITY Tissue Clearing System set at 0.8 A and 37°C for 15 hr (Logos Biosystems). After electrophoresis, the brain was washed in 50 mL of ×1 PBS three times, 1 hr each, then once overnight at room temperature. To render the tissue transparent for imaging, the brains were immersed in sRIMS (70% w/v D-sorbitol in 0.02 M sodium phosphate buffer, pH 7.4) at 4°C until transparent. The RI of the sRIMS was measured on a refractometer (Atago, PAL-RI 3850) and the RI was between 1.42 and 1.43.

### Fast 3D clearing

Fast 3D clearing was performed according to *Kosmidis et al., 2021*. Briefly, after perfusion and overnight fixation, samples were placed on an orbital shaker at 4°C, protected from light, and incubated with 20 mL of the following solutions in sequence: (1) 50% (v/v) THF prepared in sterile Milli-Q $H_2O$ with 20 μL of triethylamine (pH 9.0) (Millipore-Sigma, T0886) for 1 hr; (2) 70% THF with 30 μL of triethylamine for 1 hr; and (3) 90% THF with 60 μL of triethylamine overnight. After overnight incubation with 90% THF, samples were then rehydrated with the following solutions: (4) 70% THF with 30 μL of triethylamine for 1 hr followed by (5) 50% THF with 20 μL of triethylamine for 1 hr. Finally, samples were washed with sterile Milli-Q $H_2O$ four times, 10 min each, then washed overnight in sterilized MQ water. To prepare samples for imaging, brains were then incubated with 4 mL of Fast 3D Clear solution. To prepare the Fast 3D Clear solution, 48 g of Histodenz (Millipore-Sigma, D2158), 0.6 g of diatrizoic acid (Millipore-Sigma, D9268), 1.0 g of *N*-methyl-D-glucamine (Millipore-Sigma, M2004), 10 g of urea, and 0.008 g of sodium azide were mixed in a 100 mL beaker. Twenty mL of sterile Milli-Q $H_2O$ was added to dissolve the powder using a stir bar overnight. The final volume of the solution was approximately 50 mL. The dissolved solution was filtered and stored at room temperature. The RI of the Fast 3D clear solution was measured on a refractometer (Atago, PAL-RI 3850) and the RI was between 1.511 and 1.513. The sample was protected from light, placed on a shaker, and rocked gently at room temperature for 24 hr to render the sample transparent.

## Sample mounting and whole brain lightsheet imaging

A custom-designed sample holder was used for mounting the sample and imaging on a Zeiss Lightsheet Z.1. The sample holder consisted of a 1-inch plastic slotted screw (McMaster-Carr, 94690A724) with a nylon hex nut (McMaster-Carr, 94812A200), a custom-made magnetic nut, and a 1/8" washer (*Figure 7A*). To mount samples, we first applied a small amount of superglue gel (Loctite Gel Control) to the surface of the screw head, and gently pressed the screw head to the brain stem in order to attach it (*Figure 7B*). Then, the luer-lock tip of a 5 mL syringe (BD Bioscience, 309646) was removed using a razor blade, and the remaining syringe was used as a casting mold for mounting the brain. The brain was immersed in 3 mL of 1% melted agarose prepared in sterile water, and then the solution and immersed brain were gently aspirated into the sample holder until the agarose solidified completely (*Figure 7C*). Next, the plunger of the syringe was gently pressed to extend the agarose cylinder containing the brain out onto a 10 cm Petri dish (*Figure 7D*). Extra solidified agarose at the bottom of the cylinder was trimmed off and then the embedded sample and holder were immersed in 25 mL of EZ View imaging solution inside a 50 mL conical tube and this tube was rocked gently in a vertical position on a horizontal orbital shaker at room temperature overnight to equilibrate the sample.

To image the samples on the Zeiss Lightsheet Z.1, a custom-made, enlarged sample chamber was created, which allows sample diameters up to 1 cm compared to the original Zeiss ×5 chamber which only accommodates samples up to 4 mm (*Figure 7E–G*). A 1 mL syringe with a recessed magnet was then used as sample probe to attach to the sample holder (*Figure 7H, I*). To mount the sample on Lightsheet Z.1 for imaging, the 1 mL syringe with a recessed magnet was first loaded onto the system. The sample was placed into the custom-designed imaging chamber by hanging the mounted sample with a PTFE chamber cover with 3 mm opening (Zeiss) by the magnetic nut (*Figure 7J*). The imaging chamber with the hanging sample was assembled onto the lightsheet chamber (*Figure 7K*). The sample was then attached to the probe by lifting the sample with the chamber cover and attaching the magnetic nut to the probe (*Figure 7L and M*). The PTFE chamber cover was removed prior to imaging with a ×5/0.16 air detection lens and ×5/0.1 illumination lens. The data was acquired with a ×0.5 zoom at a resolution of 1.829 µm × 1.829 µm × 3.627 µm (X:Y:Z) in a tiled sequence with 20% overlap between tiled images. Acquired tiles were aligned and stitched together using Arivis Vision4D.

## Wholemount immunofluorescence staining of EZ Clear processed mouse brain

Mouse brains processed with the EZ Clear lipid removal and washing steps were used for standard immunofluorescent staining or iDISCO staining. For standard immunofluorescent staining, EZ Clear processed mouse brains were immersed in blocking buffer (×1 PBS + 0.08% Triton X-100 + 2% donkey serum + 0.05% sodium azide) at room temperature for 2 days, followed by incubation with primary antibodies (GFAP at 1:200 or αSMA-Cy3 at 1:200) diluted in 4 mL of blocking buffer for 4 days at room temperature. Whole brains were washed three times, 2 hr each in ×1 PBS, then once more in ×1 PBS overnight at room temperature. The samples were then incubated with secondary antibodies (1:500) and To-Pro 3 (1:1000) in blocking buffer for 4 days at room temperature with gentle agitation on an orbital shaker. The brains were then washed three times, 2 hr each wash, in ×1 PBS, then once more in ×1 PBS overnight at room temperature. The samples were then incubated in EZ View solution at room temperature for 24 hr to render the sample transparent.

For staining whole brains with the iDISCO protocol (*Renier et al., 2014*), EZ Clear processed mouse brains were incubated in permeabilization solution (×1 PBS + 0.2% Triton X-100 + 20% DMSO+ 0.3 M glycine + 0.05% sodium azide) at room temperature for 2 days, then incubated in iDISCO's blocking solution (×1 PBS + 0.2% Triton X-100 +10% DMSO+ 6% donkey serum) at room temperature for 2 days. Samples were then washed in PTwH (×1 PBS + 0.2% Tween-20 with 10 µg/mL heparin) for 1 hr, twice, then incubated with primary antibody (GFAP at 1:200, αSMA-Cy3 at 1:200) in primary antibody incubation solution (PTwH + 5% DMSO+ 3% donkey serum) at room temperature for 4 days. Samples were then washed in PTwH four to five times at room temperature until the following day, then incubated with secondary antibody (1:500) and To-Pro-3 (1:1000) in incubation solution (PTwH + 3% donkey serum) at room temperature for 4 days, then washed in PTwH three times, 2 hr each wash in ×1 PBS, then once more in ×1 PBS overnight at room temperature, then incubated in ×1 PBS with 0.05% sodium azide for another 24 hr at room temperature. The samples were then incubated in EZ View solution at room temperature for 24 hr to render them transparent.

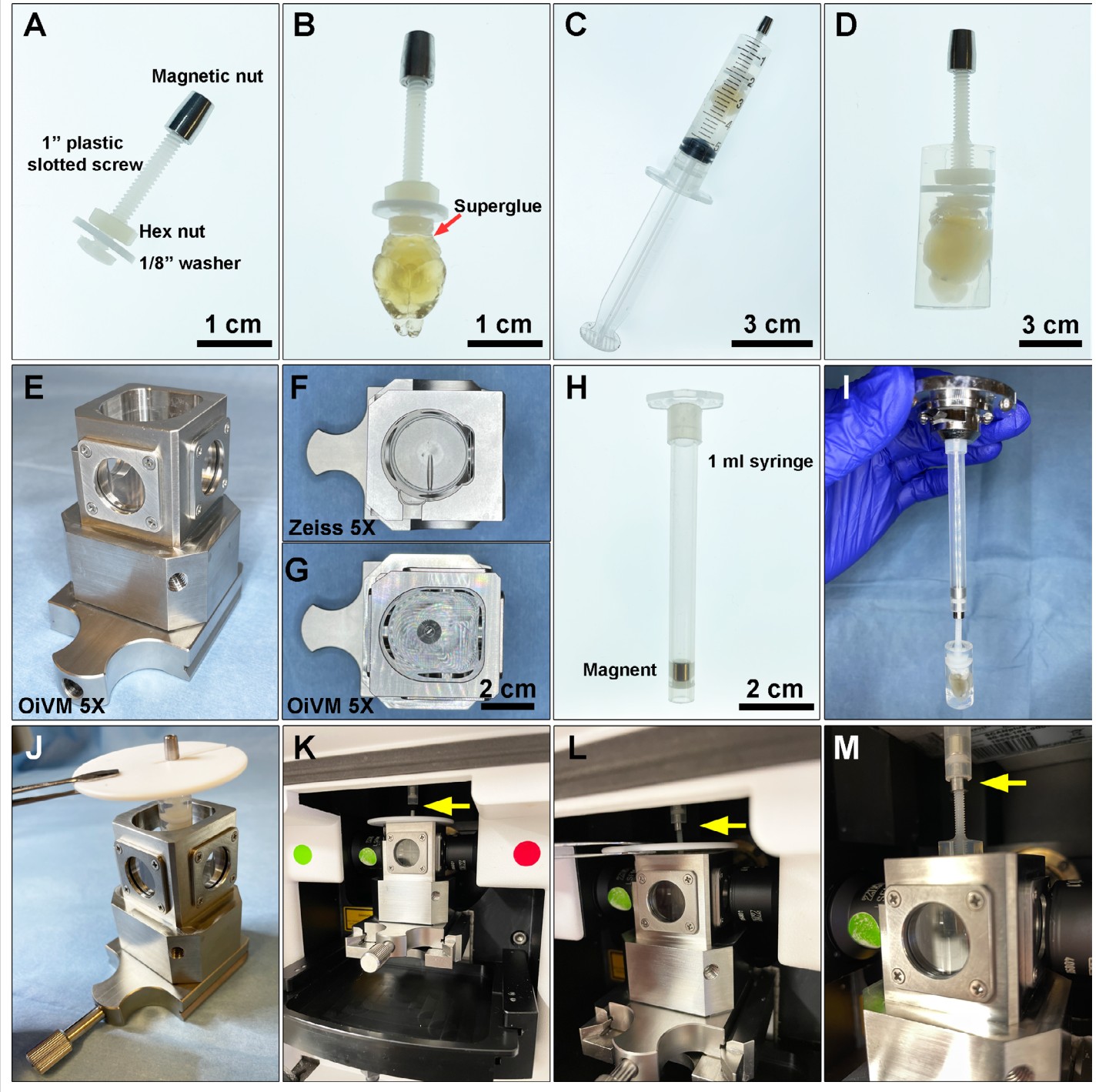

**Figure 7.** Preparing a cleared mouse brain for imaging on the Zeiss Lightsheet Z.1 platform. (**A**) A custom-designed sample holder with a magnetic nut. (**B**) A cleared brain sample was attached to the head of the plastic screw using superglue. (**C**) A 5 mL syringe is used as a casting mold for embedding the cleared brain and the tip of the sample holder in melted 1% agarose. (**D**) A cleared brain and sample holder embedded in 1% agarose. (**E**) Custom-made Optical Imaging and Vital Microscopy core (OiVM) ×5 lightsheet chamber. Comparison of the imaging chambers between (**F**) Zeiss ×5 and (**G**) OiVM ×5. (**H**) Sample mounting probe with a recessed magnet. (**I**) Demonstration of how the sample mounting probe and sample holder attach. To load the sample into the lightsheet Z.1 chamber, first (**J**) the embedded sample and holder are placed in the custom-designed imaging chamber by hanging with a PTFE chamber cover with 3 mm opening (Zeiss) by the magnetic nut, then (**K**) the imaging chamber with sample hanging was assembled onto lightsheet. (**L and M**) The sample holder was then attached to the probe by lifting the sample, the cover, and attached the magnetic nut to the magnet in the probe (yellow arrow). The chamber cover is removed and then the sample may be imaged.

## Preparation of EZ Clear processed and imaged samples for cryosectioning

After whole brains were imaged in the EZ View solution, samples were equilibrated in PBS by washing in 50 mL of ×1 PBS four times, 1 hr each, and then once more overnight at room temperature. The brains were then immersed in a three-step sucrose gradient (10%, 20%, and 30%, prepared in ×1 PBS). The samples were then incubated at 4°C for each step until the tissue sank to the bottom of the vial overnight at 4°C. Samples were then embedded in optimal cutting temperature (OCT) medium (Sakura, 4583) and snap-frozen on a bed of crushed dry ice, then stored at –80°C until ready for processing. The frozen block was sectioned on a cryostat (Leica) at 12 µm for H&E staining or at 100 µm for immunofluorescent staining.

## Histology staining

Before histological staining, slides containing mounted cryosections were equilibrated at room temperature, followed by rehydration in PBS. To perform H&E staining, slides were then incubated for 2 min in Modified Harris Hematoxylin Solution (Millipore-Sigma, HHS32-1L), rinsed with deionized water five times, then incubated for 5 min in tap water and then dipped 12 times in acidic ethanol (0.25% HCl in 70% ethanol), incubated 1 min in tap water two times, and then incubated for 2 min in deionized water. Slides were then incubated in Eosin Y Phloxine B Solution (EMS, 26051-21) for 30 s and dehydrated through an ethanol graded series (twice in 70%, twice in 80%, twice in 95%, and twice in 99.5%) for 1.5 min at each step. Lastly, slides were washed in Histoclear II (EMS, 64111-04) twice and coverslipped in DPX new (Millipore-Sigma, HX68428779) and stored at room temperature.

## Immunofluorescent staining of cryosectioned slides

One-hundred µm sections were collected in ×1 PBS with 0.05% sodium azide in a 24-well plate ('free floating'). For staining, sections were then incubated overnight in blocking buffer (×1 PBS + 0.08% Triton X-100 + 2% donkey serum) overnight at 4°C, followed by incubation with primary antibodies (CD31 at 1:200, GFAP at 1:200, beta-III tubulin at 1:200) in blocking buffer overnight at 4°C. Sections were then washed three times, 1 hr each in ×1 PBS, followed by incubation with secondary antibodies (1:500) and Hoechst (Millipore-Sigma, 14533, 10 mg/mL stock at 1:1000 dilution) in blocking buffer overnight at 4°C with gentle agitation on an orbital shaker. Sections stained with αSMA-Cy3 (1:200) were stained together with the Hoechst. The next day, after three, 1 hr washes in ×1 PBS at 4°C, sections were mounted on a charged slide (VWR Micro Slides, 48311-703) with 200 µL of EZ Clear imaging solution and coverslipped with #1 cover glass (VWR Micro Coverglass, 48366-067).

## Quantitative comparison on brain size changes, mean fluorescence intensity, and Michelson contrast

To compare the size changes of brains processed with different clearing protocols, brightfield images were captured using a Zeiss Stemi stereomicroscope and the Labscope Material App at the following stages: after perfusion, lipid removal, and RI matching. For brains that were too large to fit in a single field of view, four tiled images with at least 20% overlap were taken to cover the entire sample and stitched together using Arivis Vision4D. The size of each brain at different stages was quantified using Fiji. The 'Selection Brush Tool' under 'Oval selections' was used to outline the boundary of the brain from the captured brightfield images to measure the total pixel number within the outlined area.

To measure the fluorescence intensity and contrast at different imaging depths, as well as the preservation of the fluorescence overtime after storing samples in EZ View, Zen (Blue), and Fiji softwares were used to measure the mean ($I_{mean}$), maximum ($I_{max}$), and minimum ($I_{min}$) across the entire image at different imaging depths (z) through each mouse brains and organs imaged on lightsheet. For quantitative analysis of the signal intensity over imaging depth (lectin-649, αSMA-Cy3, and To-Pro 3), each experiment was acquired using the same imaging parameters for all wholemount samples (n=3).

To calculate the contrast of each image acquired through the imaging depth (z), the maximum ($I_{max}$) and minimum ($I_{min}$) of the perfused Lectin-649 signal of each image was used to calculate the Michelson contrast (*Wiebel et al., 2016*) and plotted against imaging depth.

$$\text{Michelson contrast} = (I_{max} - I_{min}) / (I_{max} + I_{min})$$

## Statistical analysis

Statistics analysis of ANOVA (one-way and two-way), t-test, and multiple comparisons were performed with GraphPad Prism 9 software.

# Acknowledgements

We thank Sih-Rong Wu and Dr Huda Zoghbi for providing the *Thy1-EGFP* mouse brains and Dr Nanbing Li-Villarreal for critical review of the manuscript. We thank Dr. Kristy Red-Horse (Stanford University) for providing the inspiration to adopt the Evans Blue (EB) perfusion methodology by sharing her group's beautiful data in the mouse coronary vasculature at a NAVBO meeting some years ago, and we are indebted to Dr. Lori O'Brien (University of North Carolina, Chappel Hill) for developing the EB method and sharing her detailed protocol with us for EB perfusion. This project was supported by the Optical Imaging and Vital Microscopy (OiVM) Core for all imaging experiments and the Bioengineering Core at Baylor College of Medicine for manufacturing the custom imaging chamber for the Zeiss Lightsheet Z.1. The authors also want to thank Dr Cecilia Ljungberg and the RNA In Situ Hybridization Core at Baylor College of Medicine, which is, in part, supported by a Shared Instrumentation grant from the NIH (1S10OD016167). This work was supported by grants from the National Institutes of Health (5T32GM088129-10 to WDT, R01HL146745 to MED and JDW, R01HD099026 to MED, U42OD026645 to MED, R01HL159159 to JDW), the American Heart Association (22PRE916015 to CFS), the Cancer Prevention Research Institute of Texas (RP200402 to JDW), the Department of Defense (W81XWH-18-1-0350 to JDW) and the Canadian Institutes of Health Research (PJT-155922 to JDW).

# Additional information

## Funding

| Funder | Grant reference number | Author |
| --- | --- | --- |
| Cancer Prevention and Research Institute of Texas | RP200402 | Joshua D Wythe |
| U.S. Department of Defense | W81XWH-18-1-0350 | Joshua D Wythe |
| Canadian Institutes of Health Research | PJT-155922 | Joshua D Wythe |
| American Heart Association | 22PRE916015 | Carlos P Flores Suarez |
| National Institutes of Health | 5T32GM088129-10 | Williamson D Turner |
| National Institutes of Health | R01HL146745 | Mary E Dickinson |
| National Institutes of Health | R01HD099026 | Mary E Dickinson |
| National Institutes of Health | U42OD026645 | Mary E Dickinson |
| National Institutes of Health | R01HL159159 | Joshua D Wythe |

The funders had no role in study design, data collection and interpretation, or the decision to submit the work for publication.

## Author contributions

Chih-Wei Hsu, Conceptualization, Resources, Data curation, Formal analysis, Supervision, Validation, Investigation, Visualization, Methodology, Writing – original draft, Project administration, Writing – review and editing; Juan Cerda III, Williamson D Turner, Carlos P Flores Suarez, Resources, Investigation, Methodology; Jason M Kirk, Conceptualization, Resources, Validation, Methodology; Tara L

Rasmussen, Investigation, Methodology, Writing – review and editing; Mary E Dickinson, Conceptualization, Resources, Supervision, Funding acquisition; Joshua D Wythe, Conceptualization, Resources, Supervision, Funding acquisition, Validation, Methodology, Project administration, Writing – review and editing, Writing – original draft

**Author ORCIDs**
Chih-Wei Hsu  http://orcid.org/0000-0002-9591-9567
Joshua D Wythe  http://orcid.org/0000-0002-3225-2937

**Ethics**
This study was carried out in strict accordance with the recommendations in the Guide for the Care and Use of Laboratory Animals of the National Institutes of Health. All animal research was conducted according to protocols approved by the Institutional Animal Care and Use Committee (IACUC) of Baylor College of Medicine (AN-4593, AN-8329, and AN-7731).

**Decision letter and Author response**
Decision letter https://doi.org/10.7554/eLife.77419.sa1
Author response https://doi.org/10.7554/eLife.77419.sa2

---

## Additional files

**Supplementary files**
• Transparent reporting form

**Data availability**
All data generated or analyzed during this study are included in the manuscript and supporting files. Source data files have been provided for Figure 1 to 4 and Figure 1-figure supplement 1.

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
