## [Editor Report]

The manuscript reports a new tissue clearing procedure that is faster (clearing within 48 hours), uses less hazardous chemicals, and importantly appears to result in less tissue volume change compared to other methods. The simple protocol adds further to the toolbox of tissue clearing methods and is one that is likely to be even more popular than many current methods.

---

## [Decision Letter]

**Decision letter after peer review:**

Thank you for submitting your article "EZ Clear for simple, rapid, and robust mouse whole organ clearing" for consideration by *eLife*. Your article has been reviewed by 2 peer reviewers, and the evaluation has been overseen by a Reviewing Editor and Marianne Bronner as the Senior Editor. The following individual involved in review of your submission has agreed to reveal their identity: Timothy C Cox (Reviewer #2).

Essential revisions:

1) Quantification and rigorous assessment across multiple tissue types is required to support the notion that EZClear is a major advance over existing protocols.

2) Similarities between EZClear and the existing Fast3D protocol have not been clearly portrayed and more credit should be given to the existing work. Further, additional quantitative comparisons between EZ Clear and Fast 3D/3Disco are needed.

3) In addition to the comparison across multiple tissue types noted in point #1, an extended version of the EZ Clear protocol with immunofluorescent labeling procedures in whole mouse brain tissue should be included along with quantification of fluorescent intensity as a function of depth. For data currently in the manuscript, quantification is also needed to show how the intensity and contrast of the fluorescent labeling changes as a function of depth.

4) LSFM imaging should be performed in some of the other mouse tissues to demonstrate sufficient clearing for quantification purposes.

5) New data and quantification are required to test whether fluorescence intensity is maintained, fades or increases following periods of storage in EZ View.

6) Improper style/grammar issues throughout the manuscript need to be addressed. Further, slang/commercialized terms should be avoided and functional names used instead.

*Reviewer #1 (Recommendations for the authors):*

– The manuscript writing in select portions of the Methods Section could be improved. The verb tense changes and the text appears to be copied from a lab protocol (e.g. end of "Animal Perfusion" and beginning of "EZ Clear protocol").

– The text in the graph in Figure 1B is very small and difficult to read.

*Reviewer #2 (Recommendations for the authors):*

Major issues to address:

The true novelty of this method is a little unclear given the similarity between the EZ Clear and Fast3D solutions and procedure. I feel the authors should give more credit to the Fast3D procedure, as it seems clear that the current method is derived from it: both methods use THF for delipidation which is followed by a water wash, and then the samples are submerged in similar chemicals for 'viewing'. The use of a different commercial name (Nycodenz in the EZ View solution) for the same reagent used in Fast 3D (Histodenz) makes it sound more novel than it actually is. The only apparent difference between the EZ Clear and Fast3D approaches is that two minor chemicals (diatrizoic acid and glucamine) are omitted from the Fast 3D viewing solution and a graded series of THF treatments is replaced by a single treatment. However, if these differences are indeed as significant for clearing as the brain hemisphere data imply, then it would greatly benefit the manuscript if the authors could clarify which of the differences between the two procedures confer this advantage or provide some reasonable explanation as to why they see notable differences in the clearing in the one example they show.

Although one might presume the applicability of this method to be broad, like Fast3D, this would be important to demonstrate. This could be done by looking at more diverse tissue types or more examples of samples processed side by side and imaged with the two methods. This is typically done for other papers reporting clearing methods.

The majority of the remaining issues, which appear throughout the paper, and particularly in the Methods section, relate to improper style and/or grammar issues, which can be easily addressed. It is strongly recommended that all authors proof read the manuscript during revision as this does not appear to have occurred with the original submission.

These are listed below largely in the order they appear:

1. The authors have chosen to name their method "EZ Clear", understandably for catchiness and simplicity. However, they should avoid using slang terms such as "EZ cleared" when describing the use of this protocol. Simple rewording will address this. It also seems a little unnecessary to have "commercialized" names for each of the three steps (EZ Away, EZ Wash and EZ View) versus 'functional' names: delipidation, water wash, imaging solution. This is not done for other methods unless they are already commercialized.

2. Line 51: should be 'hydrogel scaffolding-based' not 'hydrogel-scaffolding based'.

3. Line 64: 'limit its' should be 'limit their' as it is referencing plural.

4. Line 70: The comment regarding 'the post-processing of samples following organic solvent clearing for histology and immunohistochemistry are yet to be seriously explored' is not strictly true. It is somewhat limited but there are existing protocols to do this for at least BABB cleared specimens (eg. following OPT imaging). They are just not widely utilized.

5. There are numerous places within the manuscript where the English is poor, including instances where the participle 'the' or another word is missing. In addition, in multiple places throughout the Methods, there are changes in tense and person – where the methods revert to what sounds like a step-by-step lab protocol (ie. instructional vs past tense descriptive). These are basic issues in scientific manuscript writing which raise the concern expressed above about whether many of the authors actually read the entirety of the manuscript before submission.

6. The first section in the Results is titled "EZ Clear: simple, rapid, and effective", which sound like commercial brochure material rather than a section of a scientific manuscript. The remaining titles in the results are more appropriate.

7. There are numerous sentences within the Results that do not belong in the Results but rather either in the Methods or in the Discussion (or essentially duplicate what is said in those sections). For example, the first 9 lines of the Results comprises general statements about the procedure as well as the composition of some of the solutions, which seem more appropriate for the Discussion and Methods, respectively.

8. On line 94, the beginning of the sentence "Compared to brains fixed in only 4% paraformaldehyde (PFA)," seems a pointless comparison and should be removed.

9. The title for Figure 2 is not informative or reflective of the full content of the figure and thus needs to be reworded.

10. On line 382, the words "before tissue was harvested" should be deleted as it is an odd/unnecessary place to state this since the following line states that the "mice were then deeply anesthetized".

11. The sentence starting on line 399 (and many subsequent places in the Methods) has altered tense relative to other sentences.

12. Line 400: please replace "XXX" with actual words.

13. It is suggested that the composition of each "EZ" solution is listed separately either in a separate section preceding "EZ Clear protocol" or as the lead paragraph to that section as this will simplified the reading of this section. That said, assigning the name 'EZ Wash' to something that is simply MilliQ water seems unnecessary.

14. Sentence beginning on line 409 is in the wrong tense.

15. Line 413: "remain" should be "remaining". "Delipided" is a descriptor (slang term?) – please use better English to describe the processed samples.

16. From line 414 needs to be reworded to fix English and tense. It is also "7M urea" not "7M of urea".

17. Line 418: what 'phosphate' is used in the 0.02M phosphate buffer that is referred to multiple times?

18. Line 442: fix English – it should not be "was then undergo".

19. Line 474: why were measurements done on stitched together 2D images of the brains versus post-scanning where the actual volume can be calculated by any number of software.

20. Line 496: sentences have a 'protocol' type format/structure -change to standard format.

21. Line 505 – section title should be changed to "Preparation of cleared samples for cryo-sectioning".

The actual quality of the data that is presented is very good. It is just the true novelty, impact and conclusions (and writing style/English) that need to be tightened up.

---

## [Author Response]

Essential revisions:1) Quantification and rigorous assessment across multiple tissue types is required to support the notion that EZClear is a major advance over existing protocols.

We thank the reviewers for their comments regarding the addition of quantitative assessment across multiple tissue types. We have expanded our analysis of cleared tissue types to include murine brain, eye, heart, kidney, testis, and ovary with endothelium labeled via perfused lectin649 and included the results in Figure 1. All organs shown in the revised manuscript were processed using the same EZ Clear procedure (3 steps in 48 hours) and can be imaged through their entire volume with no significant difference in contrast across the various imaging depths, as shown in Figure 3H and 3I.

We have also quantitatively compared the difference between imaging tissues in RIMS and EZ View by measuring the mean fluorescence intensity through the depth of the tissue to show that while samples equilibrated in EZ View maintain fluorescence intensity throughout various imaging depths (0 to 5 mm), the detectable fluorescence intensity of samples equilibrated in RIMS gradually decreased when imaging deeper depths in brain tissue, with significant difference at a depth of 4 mm when compared to samples equilibrated in EZ View (highlighted in revised Figure 2K).

2) Similarities between EZClear and the existing Fast3D protocol have not been clearly portrayed and more credit should be given to the existing work. Further, additional quantitative comparisons between EZ Clear and Fast 3D/3Disco are needed.

While both EZ Clear and Fast 3D use a similar strategy of removing lipids by incubating in THF, and they both contain urea in the imaging media, the development of EZ Clear is completely independent of Fast 3D. Our rational for using 50% THF for removing lipids while keeping samples hydrated and using urea to adjust the refractive index of the nycodenz solution are novel and did not derivefrom Fast 3D. However, we acknowledge that Fast 3D was published before our work (Nov 22, 2019 vs our posting on BioRxiv on Jan 13, 2022) and more credit should be given. We have included a section in the discussion to clarify and more accurately describe the similarities between EZ Clear and Fast 3D. In addition, we have revised Figure 3 to include a quantitative comparison of how fluorescence intensity and contrast changes as a function of depth between samples processed with EZ Clear and Fast 3D. We did not include the samples processed with 3DISCO because our Lightsheet Z.1 system is not compatible with imaging samples RI-matched with organic solvents, which is also one of the reasons we want to develop EZ Clear.

3) In addition to the comparison across multiple tissue types noted in point #1, an extended version of the EZ Clear protocol with immunofluorescent labeling procedures in whole mouse brain tissue should be included along with quantification of fluorescent intensity as a function of depth. For data currently in the manuscript, quantification is also needed to show how the intensity and contrast of the fluorescent labeling changes as a function of depth.

We thank the reviewer for the suggestion to extend our EZ Clear protocol to accommodate wholemount immunolabeling. The revised manuscript now includes a wholemount immunostaining protocol validated using adult mouse brains processed with EZ Clear then stained using a standard immunostaining procedure or processed using an IHC procedure adapted from iDISCO (Renier et al., Cell. 2014). In the revised manuscript we quantitatively compare these stained samples to show that not only do the antibodies diffuse through the brains efficiently with both staining protocols (measured laterally and axially), but there are no detectable differences between either protocol in terms of changing in intensity as a function of depth.

4) LSFM imaging should be performed in some of the other mouse tissues to demonstrate sufficient clearing for quantification purposes.

We appreciate this suggestion. Revised Figure 1 now includes LSFM imaging of murine brain, eye, heart, kidney, testis, and ovary processed with EZ Clear.

5) New data and quantification are required to test whether fluorescence intensity is maintained, fades or increases following periods of storage in EZ View.

In the revised manuscript we imaged samples stored in EZ View for 70 days and found that the fluorescence intensity of perfused lectin-649 is maintained, suggesting EZ View is compatible with long-term sample storage (Figure 2L)

6) Improper style/grammar issues throughout the manuscript need to be addressed. Further, slang/commercialized terms should be avoided and functional names used instead.

We have carefully edited the resubmission and corrected improper style, grammar, and inappropriate syntax. We have also changed the slang/commercial terms (EZ Away/EZ Wash/EZ View) for the 3 steps clearing procedure and use functional names (lipid removal/ washing/ RI matching) instead.